# Caspar specifies primordial germ cell count and identity in *Drosophila melanogaster*

**Subhradip Das[1†], Sushmitha Hegde[1†], Neel Wagh[1], Jyothish Sudhakaran[1], Adheena Elsa Roy[1], Girish Deshpande[1,2]\*, Girish S Ratnaparkhi[1]\***

[1]Department of Biology, Indian Institute of Science Education & Research, Pune, India; [2]Department of Molecular Biology, Princeton University, Princeton, United States

**\*For correspondence:**
gdeshpande@princeton.edu (GD);

girish@iiserpune.ac.in (GSR)

†These authors contributed equally to this work

**Competing interest:** The authors declare that no competing interests exist.

## eLife Assessment

This study investigates the role of Caspar (Casp), an orthologue of human Fas-associated factor-1, in regulating the number of primordial germ cells that form during *Drosophila* embryogenesis. The findings are **important** in that they reveal an additional pathway that contributes to germ cell specification and maintenance. The evidence supporting the conclusions is **solid**, as the authors identify Casp and its binding partner Transitional endoplasmic reticulum 94 (TER94) as factors that influence germ cell numbers.

## Abstract

Repurposing of pleiotropic factors during execution of diverse cellular processes has emerged as a regulatory paradigm. Embryonic development in metazoans is controlled by maternal factors deposited in the egg during oogenesis. Here, we explore maternal role(s) of Caspar (Casp), the *Drosophila* orthologue of human Fas-associated factor-1 (FAF1) originally implicated in host-defense as a negative regulator of NF-κB signaling. Maternal loss of either Casp or it's protein partner, transitional endoplasmic reticulum 94 (TER94) leads to partial embryonic lethality correlated with aberrant centrosome behavior, cytoskeletal abnormalities, and defective gastrulation. Although ubiquitously distributed, both proteins are enriched in the primordial germ cells (PGCs), and in keeping with the centrosome problems, mutant embryos display a significant reduction in the PGC count. Moreover, the total number of pole buds is directly proportional to the level of Casp. Consistently, it's 'loss' and 'gain' results in respective reduction and increase in the Oskar protein levels, the master determinant of PGC fate. To elucidate this regulatory loop, we analyzed several known components of mid-blastula transition and identify the translational repressor Smaug, a zygotic regulator of germ cell specification, as a potential critical target. We present a detailed structure-function analysis of Casp aimed at understanding its novel involvement during PGC development.

## Introduction

As a model organism, *Drosophila melanogaster*, has been instrumental in establishing and advancing several developmental paradigms underlying embryonic development. *Drosophila* has also emerged as a relatively simple yet tremendously useful model, to analyze the underpinnings of the immune response (*Medzhitov, 2001*; *Lemaitre and Hoffmann, 2007*). Critically, despite the small size and modest cellular complexity, the insect immune system shares many fundamental traits with the higher vertebrates, including the humoral and innate arms and the dialogue between the two (*Hultmark, 1993*; *Adams et al., 2000*). Furthermore, several factors that contribute to the proper functioning

of the insect immune system are highly conserved and perform analogous functions across evolution (*Hoffmann and Reichhart, 2002*; *Lemaitre and Hoffmann, 2007*).

Interestingly, almost all the protein components of the insect immune system are highly pleiotropic and involve diverse activities during developmental progression. For example, the Toll class of proteins belonging to a larger family of pattern recognition receptors are essential while mounting robust immune response against the microorganismal invasion (*Brennan and Anderson, 2004*; *Wang and Ligoxygakis, 2006*). The same proteins are also deployed earlier during embryonic patterning and morphogenesis (*Govind, 1999*; *Hoffmann and Reichhart, 2002*; *Valanne et al., 2011*). Evidently, in several instances, different pathway components or modules used during early development are repurposed to mediate immunity both in the insect and the mammalian context (*Belvin and Anderson, 1996*; *Govind, 1999*; *Roth, 2023*).

The involvement of early embryonic morphogen, Dorsal, during humoral response in flies is a canonical example of the context-specific and diverse functions of immune system components (*Govind, 1999*; *Roth, 2023*). Maternal loss of function of genes involved in Toll/Dorsal signaling affect Dorsoventral patterning (*Belvin and Anderson, 1996*). A nuclear-cytoplasmic gradient of Dorsal, set up by asymmetric Toll signalling establishes cell fate across the Dorsal-ventral axis in the syncytial embryo (*Nüsslein-Volhard, 2022*). Pathogenic invasion of insects induces both the humoral and cellular immune response (*Belvin and Anderson, 1996*; *Williams, 2007*). The humoral response consists of the production of antimicrobial peptides by the fat bodies, which serve as a first line of defence (*Belvin and Anderson, 1996*; *Imler and Hoffmann, 2000*). Two Rel family member proteins, Dorsal and Dif, homologous to the mammalian NF-kappa B, induce the expression of defense peptides (*Belvin and Anderson, 1996*; *Govind, 1999*; *Buchon et al., 2014*). NF-kappa B is essential for differentiating lymphocytes, which engineer the acute-phase response. Altogether, undertaking functional analysis of proteins, in a temporally distinct developmental context, has proven to be highly informative and insightful. Notably, however, thus far, such analysis has focused on only activator proteins. Here, we have investigated the embryonic function of Caspar (Casp), a protein involved in inhibiting the immune response (*Kim et al., 2006*).

Casp is an intracellular negative regulator discovered in a genetic screen to identify suppressors of antibacterial immunity (*Kim et al., 2006*). Flies mutant for *casp* were identified due to their high rates of melanization, an innate immune response that leads to encapsulation of pathogens in the gut and fat body. Interestingly, *casp* mutants isolated in this study were resistant to Gram-negative bacterial infections due to elevated expression of the antimicrobial peptide (AMP) diptericin; consequently, infected flies survived longer than their wild-type counterparts. Strikingly, *casp* overexpression inhibited nuclear localization of Rel in response to infection in the fat body. Excess levels of Casp result in the cytoplasmic retention of Rel in its uncleaved, inactive form, presumably due to inhibition of the protease, Dredd (*Kim et al., 2006*).

Intriguingly, sequence analysis of Casp indicated a high degree of similarity to the mammalian Fas-associated factor 1 (FAF1) protein (*Kim et al., 2006*; *Tendulkar et al., 2022*; *Figure 1A*). FAF1 is evolutionarily conserved and was initially discovered as an interactor of Fas, a pro-apoptotic member of the tumor necrosis factor receptor family (*Chu et al., 1995*). FAF1 also interacts with the components of the death-inducing signaling complex (DISC) such as the Fas-associated death domain (FADD) and Caspase-8 proteins (*Ryu et al., 2003*). These interactions are mediated by the death effector domains (DED) in FADD and Caspase-8 and the DED-interacting domain (DEDID) in FAF1 (*Ryu et al., 2003*). Both 'loss' and 'gain' of function experiments indicated that FAF1 is crucial in promoting cell death via transduction of the apoptotic signal (*Ryu and Kim, 2001*; *De Zio et al., 2008*). FAF1, like its *Drosophila* ortholog, is a negative regulator of NFκB signalling (*Min-Young et al., 2004*; *Park et al., 2007*).

FAF1's myriad cellular functions can be attributed to its multiple protein-interaction domains that allow its participation in ubiquitin-related processes such as protein degradation (*Song et al., 2005*; *Menges et al., 2009*; *Zhang et al., 2011*). Consistently, FAF1 harbors a Ubiquitin-associated (UBA) domain, a UAS domain and a Ubiquitin-like regulatory X (UBX) domain (*Figure 1A*). Furthermore, the N-terminal UBA domain recruits polyubiquitinated proteins, leading to their accumulation (*Song et al., 2009*). The C-terminal UBX domain, on the other hand, interacts with the molecular chaperone, valosin-containing protein (VCP/p97) bound to the Npl4-Ufd1 heterodimeric complex (*Schuberth and Buchberger, 2008*; *Kloppsteck et al., 2012*; *Ewens et al., 2014*). FAF1 complexed with

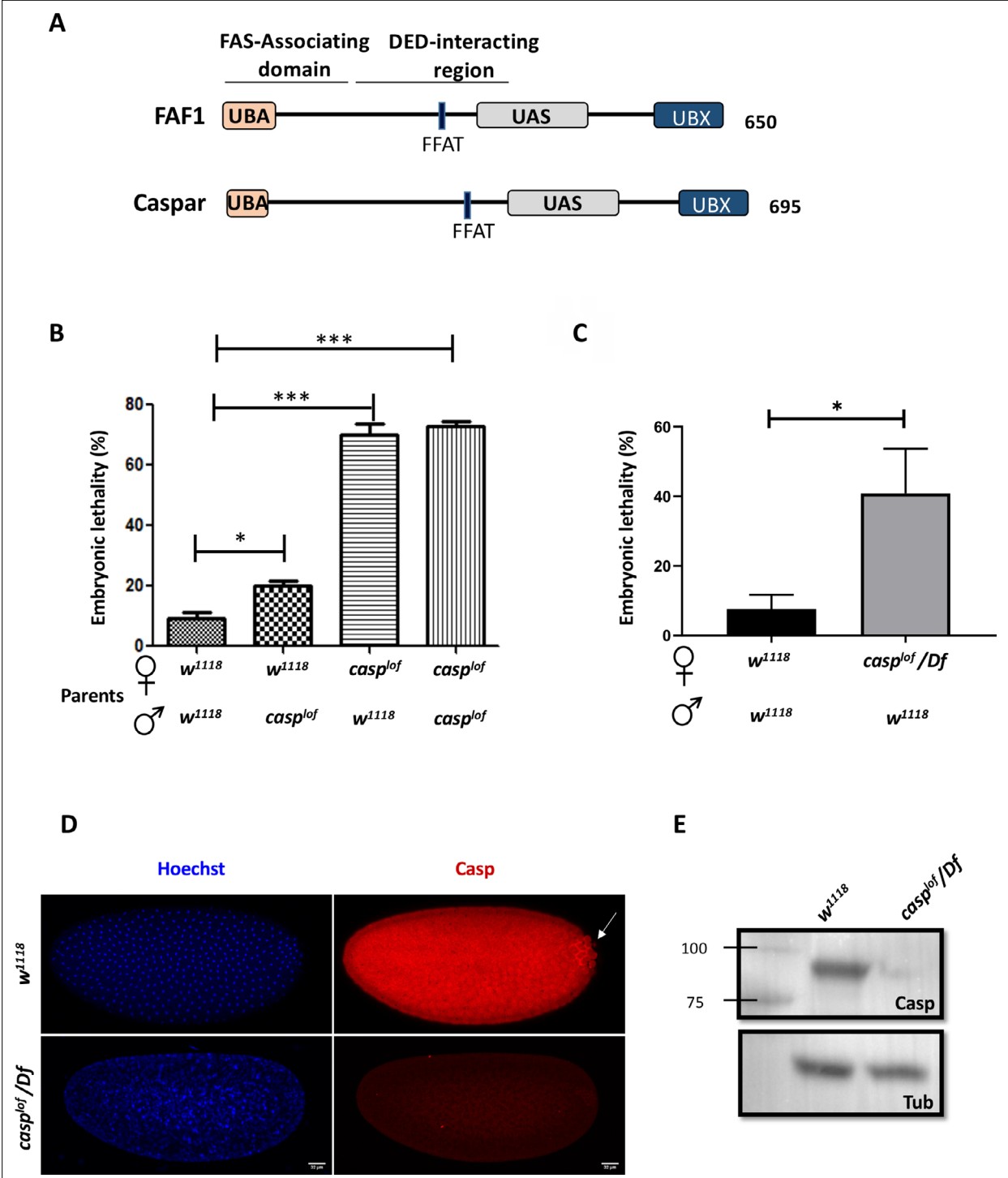

**Figure 1.** *casp* is a maternal effect gene. (**A**) Comparison between human FAF1 and Casp shows conserved protein domains, which are described in the text. (**B**) Embryos laid by homozygous *casp^{lof}* females show ~70% lethality, irrespective of the paternal genotype (*w^{1118}* or *casp^{lof}*), suggesting a strictly maternal function of *casp* (**C**) The use of a deficiency in the *casp* locus validates the lethal phenotype associated with the mutation, which drops to ~40%. In panels B and C, the parental genotype is listed on the X-axis, with percent embryonic lethality plotted as a bar graph. N=3, ordinary one-way ANOVA/ unpaired t-test, (***) p<0.001, (*) p<0.05. (**D**) Immunofluorescence images of 0–3 hr embryos derived from *w^{1118}* and *casp^{lof}/Df* females, stained with Hoechst and Casp (**E**) Casp protein levels were assessed in 0–3 embryos laid by *w^{1118}* and *Casp^{lof}/Df* animals, evaluated via western blotting. Tubulin is used as a loading control.

The online version of this article includes the following video, source data, and figure supplement(s) for figure 1:

*Figure 1 continued on next page*

*Figure 1 continued*

**Source data 1.** PDF file containing original western blots for *Figure 1E*, indicating the relevant bands.

**Source data 2.** Original unedited blots for western blot analysis displayed in *Figure 1E*.

**Figure supplement 1.** Caspar mRNA is deposited maternally in the oocyte.

**Figure supplement 2.** Time-lapse images are presented for the videos (*Figure 1—videos 1–3*) of *w*<sup>1118</sup> and *casp* mutants.

**Figure 1—video 1.** Time lapse imaging of live embryos.

https://elifesciences.org/articles/98584/figures#fig1video1

**Figure 1—video 2.** Time lapse imaging of live embryos.

https://elifesciences.org/articles/98584/figures#fig1video2

**Figure 1—video 3.** Time lapse imaging of live embryos.

https://elifesciences.org/articles/98584/figures#fig1video3

VCP-Npl4-Ufd1 and polyubiquitinated proteins is known to assist endoplasmic reticulum-associated degradation (ERAD; *Lee et al., 2013*). The UAS domain, a domain of unknown function, is involved in interaction with long-chain fatty acids, which is thought to promote the polymerization of FAF1 (*Kim et al., 2013*).

While Casp has a well-established role in the immune response, modENCODE RNAseq and proteomics data suggest that *casp* is also highly expressed in the 0–3 hr old embryo (*Brown et al., 2014*; *Casas-Vila et al., 2017*). Consistently, a snapshot of *casp* staining in a high-throughput RNA in situ experiment indicates ubiquitous expression of maternally deposited *casp* in the *Drosophila* embryo (*Weiszmann et al., 2009*). We thus wondered if Casp expression in early embryos is functionally relevant for proper developmental progression. To explore the possible developmental function of Casp, we first assessed if Casp function is needed for viability. Analysis of a hypomorphic allele of *casp* demonstrated that Casp is indeed required maternally for embryonic development. Consequently, roughly half of the embryos maternally compromised in *casp*, fail to undergo gastrulation. Furthermore, such embryos display developmental defects including aberrant cytoskeletal network starting from early blastoderm stages. Interestingly, Casp is expressed strongly in primordial germ cells (PGCs). Consistent with the enrichment, maternal reduction of *casp* significantly affects the total number of pole cells. Here, we present an analysis of *casp* function during early embryonic development and its role in the formation and/or specification of PGCs. We show that Casp activity regulates Oskar levels and centrosome function, two critical determinants of PGC fate in *Drosophila* embryo. Upon loss of *casp*, the total amount of Oskar and Smaug changed reciprocally to influence the PGC count. Ubiquitin-based protein degradation is critically involved during early embryonic events, including the maternal-zygotic transition. We present a model explaining the involvement of Casp and its protein partner, TER94, during germ cell development, considering their influence on the clearance of Smaug, a critical regulator of maternal-to-zygotic-transition (MZT).

## Results

### Casp<sup>c04227</sup> is a loss-of-function allele of casp

To better understand embryonic function of Casp, we decided to first characterize a *casp* allele, *w*<sup>1118</sup>; *pBac(PB)casp*<sup>c04227</sup> (Bloomington stock number:11373; *Figure 1—figure supplement 1A*). This allele is induced by a piggyBac insertion in the 5' regulatory region of *CG8400/FBgn0034068/Casp* locus which is situated at 2 R, region 52D14-15 (*Thibault et al., 2004*). As summarized in *Figure 1*, our data indicate that *casp*<sup>c04227</sup> is a strong hypomorphic allele and is referred to as a 'loss of function' allele (*casp*<sup>lof</sup>) here onwards.

To better understand the function of Casp, we first decided to analyze how *casp* RNA and Casp protein are expressed during embryogenesis. Consistent with the maternal effect lethality, modENCODE (*Muers, 2011*; *Chen et al., 2014*) data suggests that *casp* mRNA is deposited maternally, being highly expressed in the 0–2 hr embryo, with a drop in expression 2–4 hr post fertilization (*Figure 1—figure supplement 1B*; *Brown et al., 2014*). Publicly available in-situ databases such as FlyAtlas, Fly-FISH, and BDGP (*Weiszmann et al., 2009*; *Tomancak et al., 2002*; *Lécuyer et al., 2007*) also confirmed maternal deposition of *casp* mRNA, with ubiquitous expression seen in stage 1–3 embryos. Fly-FISH data (*Lécuyer et al., 2007*; *Wilk et al., 2016*) further suggests that expression in

the pole cells persists while maternally deposited somatic transcripts are selectively degraded from stage 4 onwards. We immunostained 0- to 3-hr-old embryos derived from $w^{1118}$ mothers with the anti-Casp antibody (*Tendulkar et al., 2022*) and found that Casp protein was predominantly cytoplasmic, and as in the case of RNA, exhibited a relatively ubiquitous distribution in the somatic compartment (*Figure 1D*). Again, consistent with the in situ hybridization pattern, Casp-specific antibody staining appeared to be enriched in the posteriorly localized PGCs compared to the surrounding soma (*Figure 1D*, arrow). To establish the specificity of the anti-Casp antibodies in the embryonic context, we stained the embryos laid by both the $casp^{lof}$/ $casp^{lof}$ and $casp^{lof}$/Df mothers. As expected, Casp protein was nearly absent in embryos of both these genotypes suggesting that Casp protein is significantly reduced in the 0- to 3-hr-old $casp^{lof}$ mutant embryos. Analysis of the embryonic lysates using western blotting, suggested that trace amounts of Casp protein (estimated 5–10% of $w^{1118}$), were present in embryos laid by $casp^{lof}$/$casp^{lof}$ and $casp^{lof}$/Df mothers, supporting the conclusion that $casp^{lof}$ is a strong hypomorphic allele and not a null (*Figure 1E*). Interestingly, mature egg chambers of wild-type animals had Casp expression in somatically derived follicle cells, but Casp protein could not be readily detected in the egg itself (*Figure 1—figure supplement 1C*). This observation suggested that, unlike the RNA, only trace amounts of Casp protein may be maternally deposited. Thus, the protein detected in the young embryos is likely generated post-fertilization, by the translation of maternally deposited *casp* mRNA.

## Casp function is required for early embryonic development

Larval cuticle patterns are an excellent readout for major patterning defects in *Drosophila* (*Nüsslein-Volhard and Wieschaus, 1980*). To better understand the function of Casp protein during embryonic development, we analyzed the cuticles of *casp* mutant embryos. 0- to 3-hr-old embryos laid by $casp^{lof}$/Df mothers were collected and aged for 22 hr to prepare cuticles (see Materials and methods). The cuticular patterns were visualized under a dark field microscope. While ~60% of larvae displayed cuticular patterns comparable to wild type larvae, development in the remaining 40% embryos appeared to have stalled before they could deposit cuticle. These data were consistent with the extent of maternal effect lethality. Next, to define the stage of developmental arrest, we performed time-lapse live Bright-field imaging (*Cavey and Lecuit, 2008*). These data (Movies, *Figure 1—videos 1–3*) confirmed that 40% of embryos did not proceed to gastrulation and showed developmental arrest before germ band extension (around stage 6; *Figure 1—figure supplement 2* B4, C4). Such embryos displayed irregular, uncoordinated morphogenetic movements with blebbing of the plasma membrane possibly inhibiting both cephalic furrow formation, and germband elongation (*Figure 1—figure supplement 2* B2, B3, C2, C3).

## Compromising casp activity leads to centrosomal abnormalities in early embryos

To trace the defects during mid-embryogenesis including failure of gastrulation, we sought to visualize $casp^{lof}$ embryos at blastoderm stage. We labelled the 0- to 3-hr-old $casp^{lof}$ and control embryos with the nuclear dye Hoechst and the cytoskeletal F-actin marker phalloidin (*Figure 2A and B*). $casp^{lof}$ embryos displayed significant structural abnormalities that are reflected in an irregular actin network that lacks stereotypical organization, which is partially interrupted in several places (*Figure 2* B1 compared to *Figure 2* A1). Similarly, unlike the age-matched control embryos, regular nuclear spacing is disrupted, and nuclei are unevenly distributed across the embryo with occasional instances of nuclear fusion and possibly mitotic asynchrony (*Figure 2B* vs *Figure 2A*).

The nuclear and cytoskeletal defects observed in *casp* embryos prompted us to analyze the centrosomes in embryos deficient in *casp* function. Centrosomes function as the microtubule organizing centers, and actomyosin-based cytoskeletal defects have been correlated with aberrant centrosome activity (*Wu and Akhmanova, 2017*; *Blake-Hedges and Megraw, 2019* and references therein). In wild-type nuclei, centrosome duplication occurs simultaneously with the initiation of the nuclear division cycle. After completion of duplication, centrosomes separate and migrate along the nuclear membrane to reach the opposite sides of the nucleus (reviewed by *Lattao et al., 2017*; *Wu and Akhmanova, 2017*; *Blake-Hedges and Megraw, 2019*). In the embryos maternally compromised for *casp*, several characteristic aspects of centrosome behavior during mitotic divisions are altered (compare *Figure 2* B3 to *Figure 2* A3). At times, centrosomes appeared to divide even without a

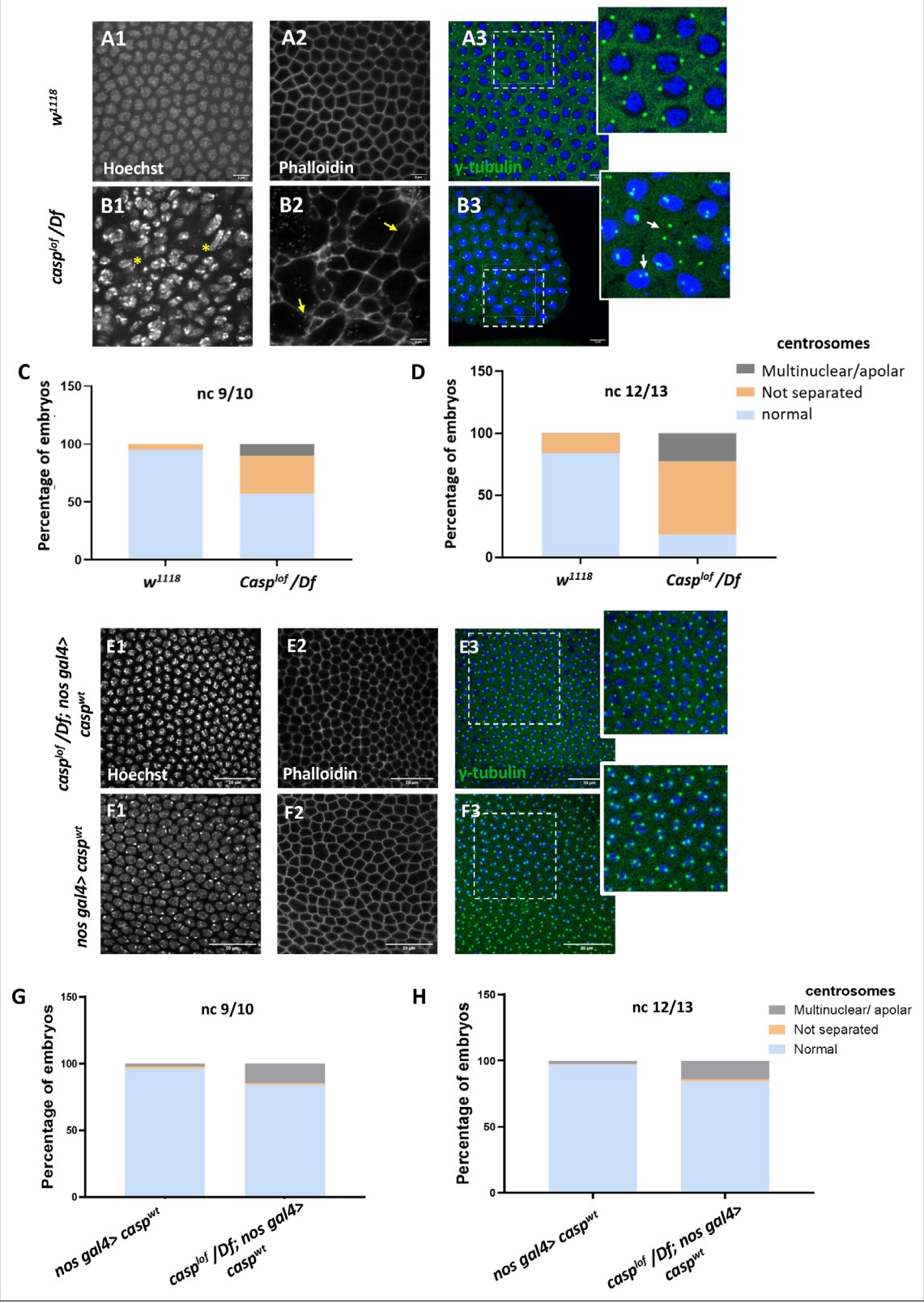

**Figure 2.** A significant proportion of embryos (~45%) laid by *casp^lof^/Df* mothers display nuclear division and cytoskeletal defects. (**A, B**) Confocal images of nuclear cycle 13/14 embryos (single sections) of the indicated genotypes stained with Hoechst, phalloidin, and gamma-tubulin. Both the regular arrangement and uniform density of nuclei are disrupted in the mutants, as indicated by a yellow asterisk. F-actin, marked with phalloidin, shows a regular, hexagonal compartmentalization in *w^1118^* (panel A2), while disorganized F-actin (white arrows; panel B3) is observed in the mutant. Defective

*Figure 2 continued on next page*

*Figure 2 continued*

centrosomes marked with gamma-tubulin can also be observed in the *Casp* mutant (inset of B3, compare to inset A3), indicated by yellow arrows. The extent of defects is quantified in nuclear cycle(nc) 9/10 (**C**) and nc 12/13 (**D**). (**E, F**) Confocal images of sections of nc 13/14 embryos of the indicated genotypes stained with Hoechst, phalloidin, and gamma-tubulin. The nuclear, cytoskeletal, and centrosomal defects are rescued (panel E) when wild-type *casp* is expressed in the *casp^lof/Df* background (compare panels **G, H** to **C, D**). Overexpression of *casp* on its own (panel F) does not appear to affect nuclear or cytoskeletal architecture. The bar charts represent the percentage of defective centrosomes in nuclear cycle 9/10 and 12/13 embryos. Number of embryos imaged ~15.

nucleus (or DNA), and many 'orphan' centrosomes devoid of nuclear DNA were observed (*Figure 2* B3, inset). By contrast, in some instances, centrosomes were duplicated but remained in proximity indicating failed or incomplete migration (see arrows in *Figure 2* B3, inset). Quantitation of the centrosomal abnormalities in *casp^lof* embryos (*Figure 2C and D*) further revealed that the behavior of the centrosomes, especially their ability to separate and migrate to the opposite poles correctly deteriorated progressively (compare *Figure 2D* to *Figure 2C*), as nuclear division cycles advanced.

To confirm the specificity of the phenotypic consequences induced by the maternal loss of *casp*, we overexpressed *UAS-casp^wt* transgene in *casp^lof/Df* embryos using *nos Gal4*, a maternal Gal4 driver. Embryos derived from such mothers were stained using nuclear dye Hoechst, phalloidin and anti-Gamma tubulin antibodies that mark the centrosomes. As shown in *Figure 2E*, maternal overexpression of *casp* substantially rescues centrosomal and cytoskeletal abnormalities seen in *casp^lof* (compare panels E1-E3 to A1-A3). This rescue is quantitated in *Figure 2G and H* (compare with *Figure 2C and D*). The *casp^lof/Df; nos Gal4/UASp-casp^wt* showed a rescue of lethality of 20–25% as compared to 40–45% for *casp^lof/Df*.

Taken together, these data showed that Casp plays an essential role in early embryonic development in *Drosophila*, and loss of <u>casp</u> results in conspicuous nuclear and cytoskeletal defects that correlate with incomplete gastrulation movements and developmental arrest, for ~45% of embryos. Moreover, such defects do not appear to be region-specific or localized within an embryo arguing in support of ubiquitous function across the embryo for Casp protein.

## Maternal Casp protein is enriched in pole cells and controls total pole cell count in blastoderm embryos

Primordial germ cells (PGCs) are precursors of the gametes—sperm and eggs. Their specification is essential for the development of the germline, which in turn passes genetic information to the next generation. During early embryonic development, PGCs are specified and segregated from somatic cells. In *Drosophila*, the PGCs are formed in the posterior end of the embryo. As the early embryonic syncytial nuclear division cycles progress, a few nuclei and centrosomes associated with them invade posteriorly localized and anchored pole plasm ahead of the rest of the somatic nuclei (*Raff and Glover, 1989*). The precocious entry of the centrosomes results in the release and microtubule-dependent transport of pole plasm, which is sequestered in newly cellularized PGCs (*Raff and Glover, 1989*; *Lerit et al., 2017*). Thus, the PGCs or the germline stem cell precursors are set aside during early embryogenesis. Moreover, centrosome behavior and dynamics are crucial for proper cellularization and mitotic cell divisions of early germ cells (*Lerit and Gavis, 2011*; *Lerit et al., 2017*). Centrosomes, the microtubule (MT)-organizing centers, ensure the faithful segregation of germ plasm, a reservoir of germ cell determinants, into PGCs. Taken together with the observation that Casp protein is readily detectable in early PGCs, we wondered if it could influence the total pole cell count. To examine this possibility, stage 5 embryos derived from *cs* and *casp^lof/Df* mothers were immunostained for the pole cell marker Vasa and visualized under a confocal fluorescence microscope. Interestingly, while *cs* and *nos-Gal4* embryos had an average of ~30 pole cells (*Figure 3* A1-B3, for quantitation see panel F) *casp^lof/Df* showed a drastic reduction in total PGC count to ~10 (*Figure 3* C1-3, F). By contrast, stage 5 embryos derived from *nosGal4 >UAS* casp mothers, where *casp* was overexpressed (*Figure 3G*), displayed considerably elevated total number of PGCs (*Figure 3* E1-E3, F). To confirm that the loss of *casp* is specifically responsible for the reduction in the total number of PGCs, we simultaneously overexpressed *casp* in the *casp^lof/Df* mothers in a germline-specific manner, which resulted in a significant rescue (~26 pole cells; *Figure 3* D1-D3, F). Taken together, these data suggested that the total pole cell number in late syncytial /cellular blastoderm embryos specifically depends on *casp*

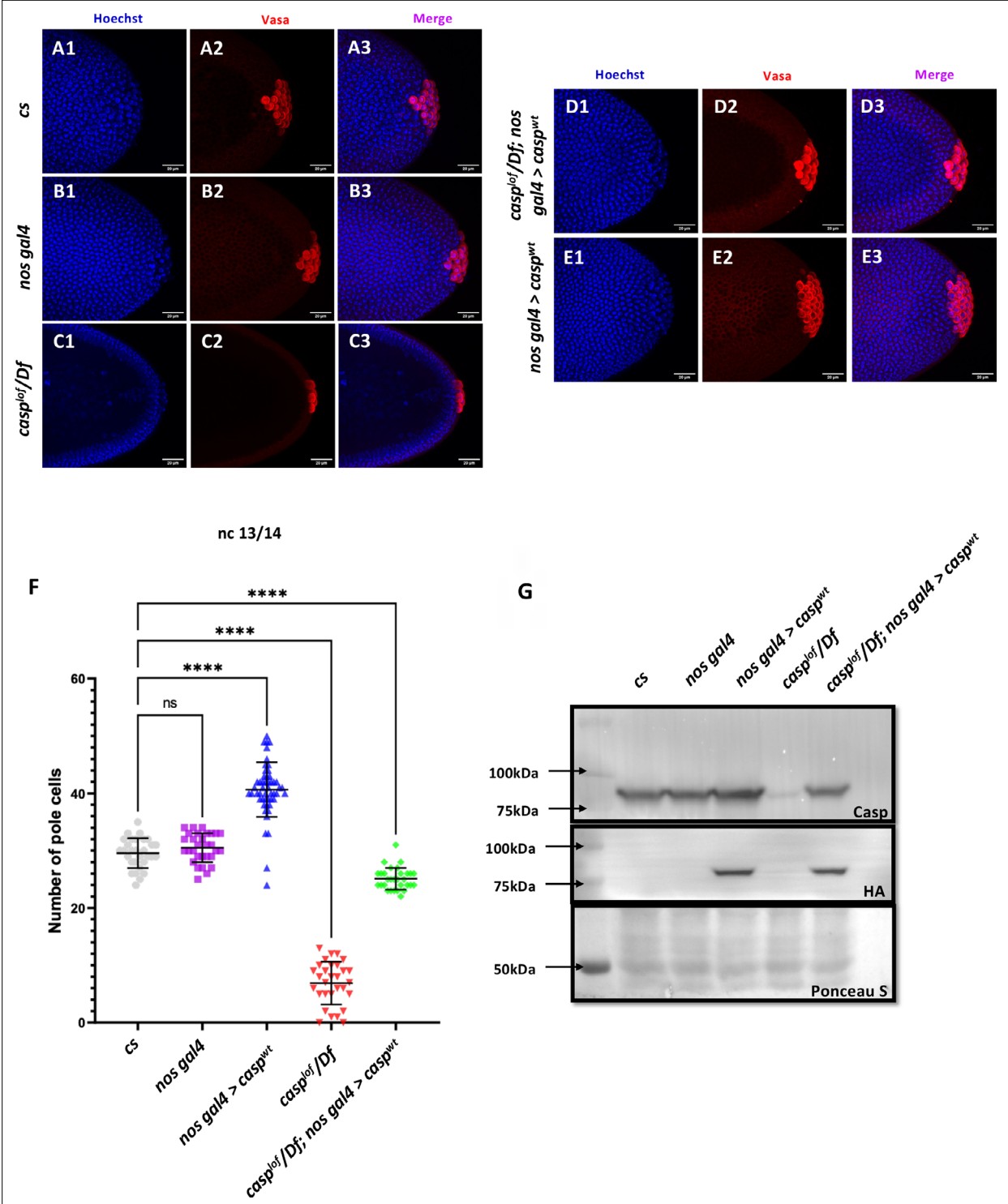

**Figure 3.** Casp influences total PGC count. (**A–E**): Shown are the Confocal images of the posterior terminii of nuclear cycle 13/14 embryos of the following genotypes: *cs* (**A1–A3**), *nos gal4* (**B1–B3**), casp$^{lof}$/Df (**C1–C3**), casp$^{lof}$/Df; nos gal4>casp$^{wt}$ (**D1–D3**) and *nos gal4>casp$^{wt}$* (**E1–E3**). Embryos were immunostained with Vasa (1:50) antibody. Hoechst marks the nuclei. (**F**) The total number of germ cells marked with Vasa were quantified and plotted as bar graphs. N (embryos)=30. Ordinary one-way ANOVA, (****) p<0.0001. (**G**) Casp protein levels were assessed in 0–3 embryos via western blotting. Rabbit anti-Casp (1:10,000) and rabbit anti-HA (1:2000) antibodies were used to probe the blot. Ponceau is used as a loading control. N=3.

The online version of this article includes the following source data and figure supplement(s) for figure 3:

**Source data 1.** PDF file containing original western blots for *Figure 3G*, indicating the relevant bands.

*Figure 3 continued on next page*

*Figure 3 continued*

**Source data 2.** Original unedited blots for western blot analysis displayed in *Figure 3G*.

**Figure supplement 1.** TER94 and germline components are major interactors of Casp.

function. Furthermore, increasing *casp* levels maternally, is sufficient to elevate the total pole cell number substantially.

## Casp interacts with TER94 in the early embryo

Data presented in the previous section demonstrated that both the 'loss' and 'gain' of *casp* activity exert reciprocal influence on total PGC numbers in early embryos. As very few germ plasm components have been shown to display this trait (*Jongens et al., 1994*), we decided to explore the phenomenon further. We have previously reported that Casp protein physically associates with both transitional endoplasmic reticulum ATPase (TER94; also called valosin-containing protein, VCP or p97) and vesicle-associated membrane protein-associated protein B (VAPB/VAP33A)(*Tendulkar et al., 2022*). Furthermore, based on interaction studies and biochemical analysis, we proposed that Casp may act as an adapter that either directly or indirectly mediates the physical association between VAPB and TER94 (*Tendulkar et al., 2022*). To arrive at this conclusion, we had primarily relied on S2 cell lysates as well as adult and fly head total protein extracts. To specifically identify the major protein partners of Casp in the early embryos, we performed similar immunoprecipitations using anti-Casp antiserum in early (0–3 hr) embryonic lysates and interactors were identified via mass spectrometry (*Figure 3—figure supplement 1A*). Peptides from 122 proteins were recovered in the Casp IP but not the IgG IP and thus, were considered significant interactors. The top 23 interactors are presented in *Figure 3—figure supplement 1A*. Interestingly, as in the case of brain and S2 cell lysates, TER94 and VAPB were the top hits, with germ cell determinants also identified as interactors (*Figure 3—figure supplement 1C*; Discussed in a subsequent section). These observations suggested that a functional protein complex between Casp, TER94, and VAPB likely exists in many different tissue/cellular contexts, including early embryos.

## Maternal requirement of TER94

Mammalian VCP/p97 is an essential chaperone for proteostasis that modulates several ubiquitin-associated processes (*Peters et al., 1990*; *Dai et al., 1998*; *Meyer et al., 2000*; *Meyer, 2005*; *Ye, 2006*; *Jentsch and Rumpf, 2007*; *Meyer et al., 2012*). Its *Drosophila* counterpart TER94 has been studied mostly in the context of neurodegeneration (*Griciuc et al., 2010*; *Chang et al., 2011*; *Azuma et al., 2014*; *Kushimura et al., 2018*; *Tendulkar et al., 2022*; *Thulasidharan et al., 2024*). Loss of TER94 was shown to ameliorate polyQ-induced eye degeneration. Moreover, the overexpression of TER94 promoted the apoptosis of neuronal cells (*Higashiyama et al., 2002*). TER94 has been uncovered in a genetic screen for maternal proteins that are phosphor-regulated (*Zhang et al., 2018*) and have roles in oogenesis and early development.

TER94 is one of the significant interactors of Casp protein in the embryonic context as suggested by the Casp interactome (*Figure 3—figure supplement 1A*) and has been implicated in ER-associated degradation (*Chang et al., 2011*; *Tendulkar et al., 2022*). Thus, we wondered if TER94 also has a maternal function and whether embryos maternally compromised for *TER94* display similar phenotypes as *casp*. To achieve this, we employed VALIUM 20/22 maternal RNAi lines (*UAS-TER94i*). First, we used a *mat-α4tubulin:VP16-Gal4* (*Mat-αtubGal4*) driver to deplete *TER94* activity in the late stages of oogenesis (*Figure 4*).

We sought to observe the consequence of *TER94* knockdown both in the adult females and early embryos derived from such females. In the females compromised for *TER94*, egg-laying behavior remained largely unaffected. Interestingly, however, the viability of such embryos was severely impaired, with >95% of embryos failing to hatch (*Figure 4A*), in agreement with earlier studies (*Zhang et al., 2018*).

A western blot of lysates derived from *Mat-GAL4 >UAS-TER94i* mothers probed with anti-TER94 antiserum indicated a robust knockdown of TER94 as compared to control (*Figure 4B*). Phenotypic analysis of embryos derived from *Mat-GAL4 >TER94* i females revealed that >70% of *Mat-GAL4 / TER94* i embryos failed to progress to the syncytial blastoderm stage, with even fewer reaching Stage

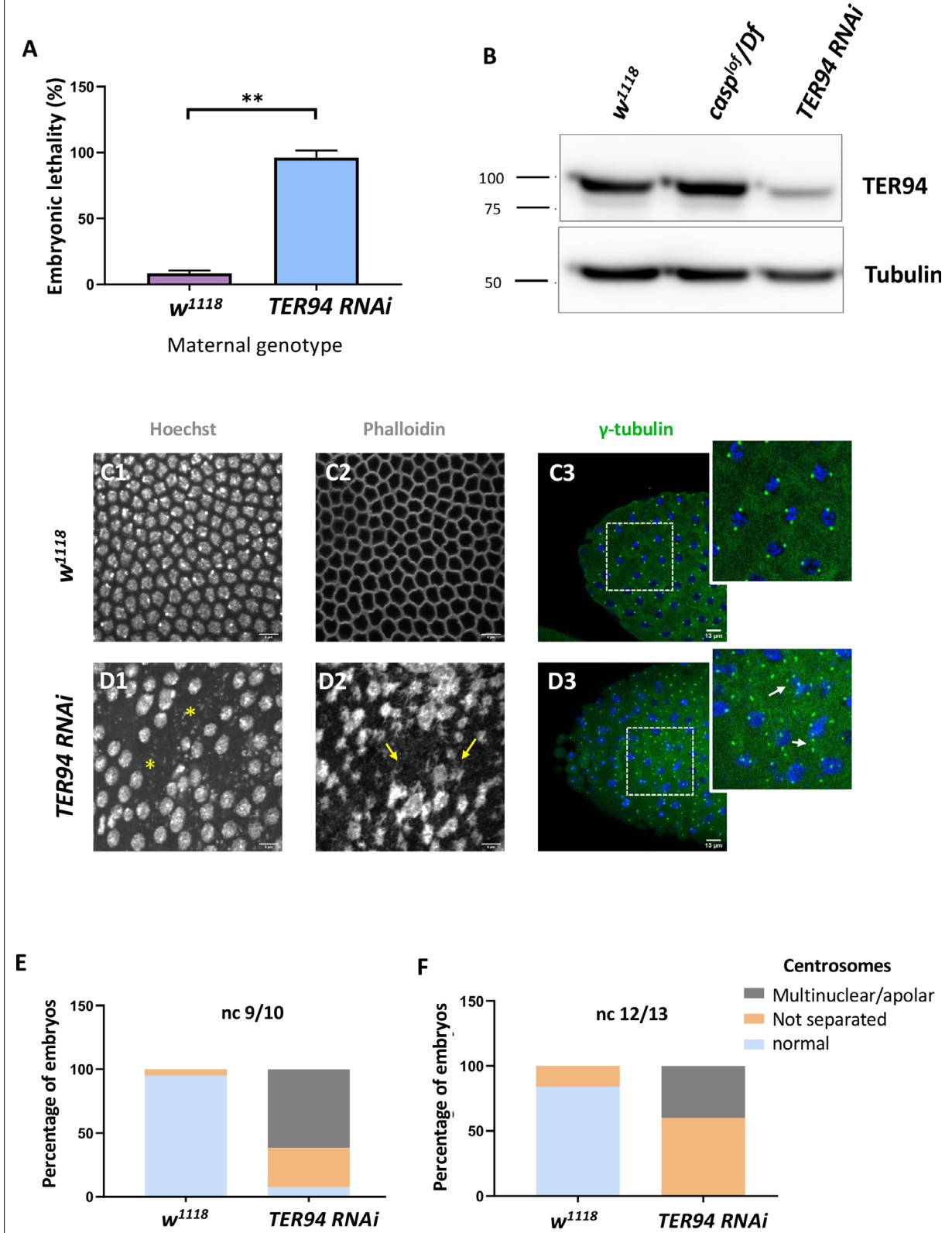

**Figure 4.** Maternal requirement of *TER94* during early embryogenesis. (**A**) Viability of embryos derived from *w^1118^* and *Mat-αtubGAL4>TER94* i (referred to as *TER94 RNAi* henceforth) mothers is represented as a bar graph. N=3, ordinary one-way ANOVA, (***) p<0.001. (**B**) Western blot analysis indicates the knockdown efficiency of TER94 in the 0–3 hr embryo, estimated to be ~90%., levels remain unaffected in *casp^lof^/Df* embryos. Tubulin was used as a loading control. N=3. (**C, D**) Confocal images of sections of nuclear cycle 13/14 embryos of the indicated genotypes stained with Hoechst, phalloidin,

*Figure 4 continued on next page*

Figure 4 continued

and gamma-tubulin. Nuclei and F-actin are visualized with Hoechst and phalloidin respectively. Nuclear disruption (D, asterisks) and F-actin aggregates (D2, arrows) are observed in the mutant. Defective centrosomes (D3, arrows, inset) can be observed in the *TER94 RNAi* embryos. (**E, F**) The percentage of defective centrosomes in nuclear cycles 9/10 and 12/13 are represented as bar charts. n=10.

The online version of this article includes the following source data for figure 4:

**Source data 1.** PDF file containing original western blots for *Figure 4B*, indicating the relevant bands.

**Source data 2.** Original unedited blots for western blot analysis displayed in *Figure 4B*.

5. The late-stage syncytial/ cellular blastoderm embryos were further assessed for cell cycle defects. As in the case of *casp^{lof}/Df*, *Mat-GAL4 >TER94* i embryos also displayed defects ranging from irregular nuclear distribution to perturbed F-actin localization, as assessed by HOECHST and phalloidin staining, respectively (compare *Figure 4* D1–D3 with *Figure 4* C1–C3).

We also examined if maternal loss of *TER94* activity results in centrosome aberrations like those observed in *casp^{lof}* embryos. Indeed, embryos deficient in *TER94* function showed both the characteristic phenotypes observed in the *casp* embryos including (a) inefficient separation of duplicated centrosomes and (b) multiple instances of 'orphan' centrosomes devoid of nuclear DNA (*Figure 4E, F*). An earlier study (*Zhang et al., 2018*) had found that TER94 RNAi leads to multi-polar spindles with supernumerary centrosomes.

Taken together, these data demonstrated that embryos maternally compromised for either Casp or TER94 functions share several phenotypes, suggesting that both the proteins likely perform essential and likely related functions during syncytial nuclear cycles in *Drosophila* embryos. Furthermore, the phenotypic consequences observed upon maternally compromising *casp* and *TER94* overlap but are not identical, with TER94 embryos showing higher penetrance in terms of phenotypes and embryonic lethality. Future experiments will be necessary to resolve the functional distinction between the two.

## TER94, a known component of pole plasm, is detectible in PGCs

Maternal loss of TER94 function mimicked both the cytoskeletal and corresponding centrosome aberrations observed upon similar loss of *casp* activity. Both centrosomes and germ plasm, are essential for the proper specification and formation of PGCs in *Drosophila* embryos (*Lerit and Gavis, 2011*; *Lerit et al., 2017*). This prompted us to examine the Casp interactome for the possible enrichment of germ plasm components and proteins that may regulate centrosome dynamics and/or behavior. *Figure 3—figure supplement 1B* lists Casp interactors that are important constituents of the germplasm, including eIF4A, me31B, and TER94, known constituents of the polar granules (*Thomson et al., 2008*).

Among the different components of the germ plasm, Oskar serves as the principal determinant of the PGC fate (*Kim-Ha et al., 1991*; *Snee and Macdonald, 2004*; *Vanzo et al., 2007*; *Lehmann, 2016*). Supporting the conclusion, the loss and gain of function of *oskar* leads to reciprocal phenotypes. Compromising *oskar* activity maternally, leads to reduction in total number of PGCs whereas anterior ectopic expression of *oskar* using the *bicoid* mRNA localization signal induces pole cell formation at the anterior (*Ephrussi and Lehmann, 1992*). The germ plasm is supplemented with mitochondria and polar granules, which consist of ribonucleoprotein complexes. Downstream of *oskar*, *vasa* and *tudor* are two genes that are essential for the assembly of pole plasm (*Breitwieser et al., 1996*; *Arkov et al., 2006*; *Jones and Macdonald, 2007*). Biochemical proteomic analysis of Vasa (VAS) and Tudor (TUD) containing polar granule complexes identified eIF4A, me31B, and TER94. This indicated that the germ plasm consists of the components of translational machinery and the endoplasmic reticulum assembly (*Thomson et al., 2008*).

TER94 is a component of pole plasm and is also physically associated with Casp. Furthermore, proteomic analysis indicated that Casp could be part of a complex comprising several germ plasm proteins including Tudor and Me31B. To confirm that, as in the case of Casp, TER94 is also detected in PGCs. We used antibodies generated against Casp (*Figure 3—figure supplement 1D*) and TER94 (*Figure 3—figure supplement 1E*) to label wild-type embryos. Embryos were also co-immunostained for germ cell specific marker, Vasa. As can be seen, both Casp and TER94 proteins are found in PGCs, colocalized with Vasa.

## Casp and TER94 regulate embryonic germ cell formation

As we were specifically interested in investigating possible similarities and distinctions between the respective functions of TER94 and Casp during early embryonic germ cell development, we decided to examine their possible functions during PGC formation (also referred to as pole cell budding). This seemed especially pertinent as centrosome behavior and dynamics are crucial for proper formation and cellularization of PGCs. Evidently, a germ plasm component, Germ-cell-less (*Cinalli and Lehmann, 2013*) was shown to be necessary for proper separation of daughter centrosomes in the dividing pole buds (*Lerit et al., 2017*). Similarly, loss of centrosome components such as Centrosomin results in partial loss of early embryonic PGCs. As maternal loss of both *casp* and *TER94* results in aberrant centrosomes in the surrounding somatic nuclei in early blastoderm embryos, we sought to investigate if posteriorly positioned centrosomes in the vicinity of pole plasm also display similar problems. Indeed, maternally compromising *casp* led to significant loss of pole buds and corresponding defective centrosome behavior (*Figure 5*).

Especially these pole buds display defective centrosome separation and/or orphan centrosomes (*Figure 5* B1–B2, D and E), as observed in the somatic nuclei. Intriguingly, while *casp^{lof}* embryos show discernible loss of pole buds, *TER94i* embryos displayed an increased number of round-shaped, 'bud-like' cells at the posterior of the stage 3 embryos (*Figure 5* C1–C2). Typically, such ectopically localized pole buds also showed inadequate separation of centrosomes and could be identified as pole buds due to the enrichment of Vasa. However, Vasa distribution and accumulation in *TER94i* buds was variable and non-uniform. Since TER94 is required for *Osk* mRNA localization in the oocyte, the defects in pole cell budding including their ectopic positioning could be a consequence of inappropriate segregation of *oskar*. Moreover, the early nuclear division cycles and nuclear migration defects seen in are defective in *TER94i* embryos could also contribute to PGC formation/ cellularization. Notably, although the pole bud count is elevated in *TER94i* embryos, presumably, these pole buds don't survive through subsequent nuclear cycles, and consequently, late blastoderm *TER94i* embryos almost completely lack PGCs.

## Casp activity is needed for the accumulation of Oskar protein at the embryonic posterior pole

Taken together, our data demonstrate that change in Casp levels leads to corresponding alteration in the total PGC count. Moreover, PGC formation is affected by maternal loss of Casp, although the precise nature of its involvement in this process remains to be determined. As Oskar is the master determinant of germ cell fate, we sought to determine if Oskar levels (*Kim-Ha et al., 1991*; *Ephrussi and Lehmann, 1992*) are correspondingly altered upon change in Casp. To assess this possibility, we stained wild-type control embryos and embryos maternally compromised for *casp* using anti-Oskar antibodies (compare *Figure 6B* to *Figure 6A*). Simultaneously, we stained embryos derived from *nosGal4 >UASp* casp mothers (*Figure 6* C1–C3) that display significantly elevated number of PGCs, presumably due to excess levels of Casp protein in the germline.

Satisfyingly, the change in Oskar protein levels is consistent with the alteration in the Casp levels in the maternal germline (*Figure 6D*). Embryos derived from *casp* mutant mothers show considerably diminished levels of Oskar as compared to the control whereas overexpression of Casp in the maternal germline results in elevation in the Oskar protein amount (*Figure 6D*). Furthermore, in all instances, Oskar protein appears to be anchored to the posterior pole as the wild-type control embryos and the only discernible change is observed in its levels.

## Casp levels influence total number of phosphor-histone3 (pH3) positive pole buds and PGCs

Changes in Casp protein levels influence the accumulation of Oskar protein. Oskar is necessary and sufficient to assemble pole plasm at the posterior pole which controls the total number of pole buds in an embryo. To directly evaluate the influence of Casp on pole bud formation, total number of pole buds and pole cells were quantitated upon maternal loss and gain of Casp against the control embryos at the same stage (*Figure 6E*). Consistent with the data presented in the previous sections, total number of pole buds (nc 9 and 10) were reduced in *casp^{lof}* embryos whereas the number was elevated in the presence of excess Casp (*Figure 6E*). Similar quantitation was performed in older embryos in two stages (nc11-12 and NC12-13), and a progressive increase was observed in the total

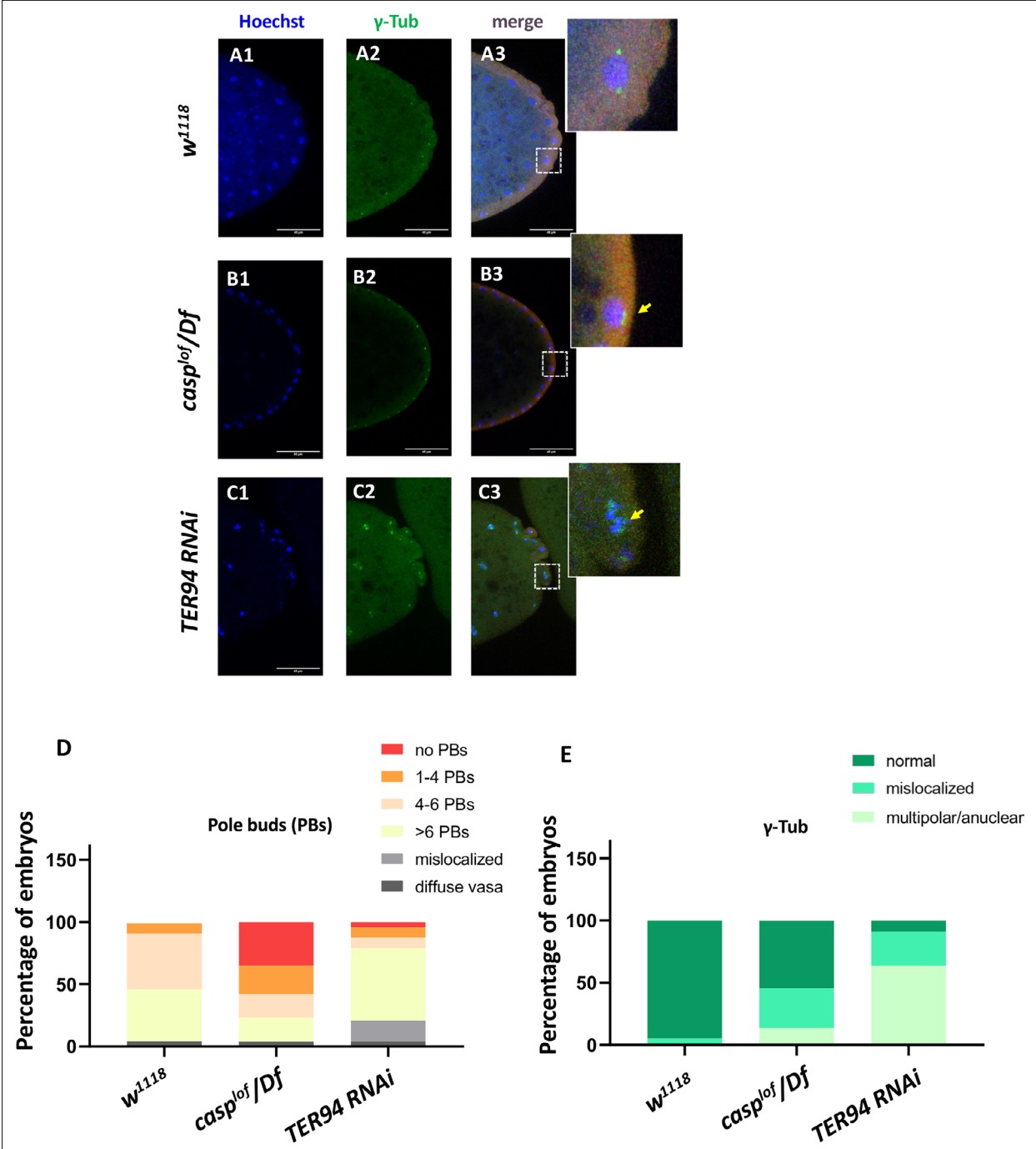

**Figure 5.** *casp and TER94 regulate pole bud formation.* (**A–C**) Confocal sections of the posterior termini of nuclear cycle 9/10 embryos derived from *w$^{1118}$* (**A1–A3**), *casp$^{lof}$/Df* (**B1–B3**), and *TER94 RNAi* (**C1–C3**) mothers immunostained with gamma-tubulin. γ-tubulin labels centrosomes of pole buds, identified based on Vasa staining. Nuclei are stained with Hoechst. The inset(s) for B3 and C3 highlight the centrosomal defects (arrow) when compared with A3. Defects were quantified and plotted as bar graphs in (**D**) and (**E**) respectively. n=20.

number of PGCs upon the gain of Casp function (***Figure 6E***). By contrast, the total PGC count for *casp$^{lof}$/Df* embryos did not increase appreciably at nc11-12 and nc12-13.

Primordial germ cell number is determined by limited mitotic cell divisions that each pole bud undergoes in a stochastic and asynchronous manner. As these are mitotic divisions, the dividing PGCs can be identified using antibodies against the miotic marker phosphoHistone3 (pH3). As the total number of pole buds and PGCs change per Casp levels, we wanted to examine if this is also reflected

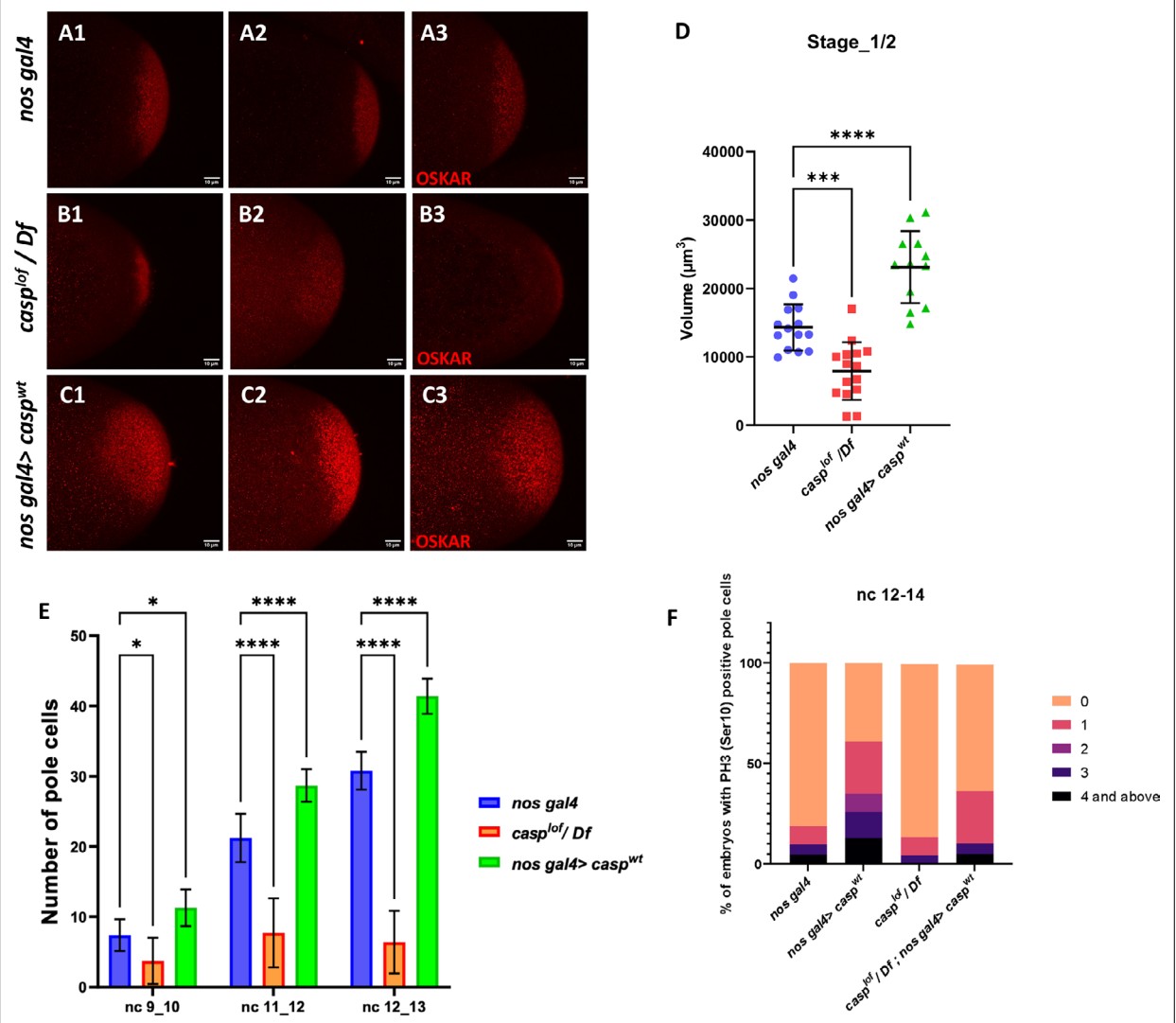

**Figure 6.** Casp regulates Oskar levels and modulates cell division. (**A–C**) Confocal microscopy images of stage 1/2 embryo laid by $w^{1118}$, $casp^{lof}/Df$, and *nos gal4>casp$^{wt}$* females and immunostained with Oskar antibodies. Replicate (1-3) are shown to highlight variable reduction in Oskar levels. In addition, as compared to controls ($w^{1118}$) embryos (**A1–A3**), spread of Oskar appears to be restricted in $casp^{lof}/Df$ (**B1–B3**) whereas it is expanded in *nos gal4>casp$^{wt}$* (**C1–C3**). (**D**) Volume occupied by Oskar, per embryo is measured and plotted for all three genotypes ($w^{1118}$, $casp^{lof}/Df$, and *nos gal4>casp$^{wt}$*), n=8. Ordinary one-way ANOVA, (**) $p<0.01$, (*) $p<0.05$. (**E**) Number of pole buds (nc 9/10) and pole cells (nc 11/12, nc 12/13) in 0- to 3-hr-old *nos gal4*, $casp^{lof}/Df$ and *nos gal4>casp$^{wt}$* embryos at different nuclear cycles. Two-way ANOVA, (*) $p<0.05$, (****) $p<0.0001$. n=10 (**F**) Graph showing the distribution of actively dividing pole cells as indicated by the presence of phospho-histone 3 (Ser10) antibody. Quantitative analysis was performed using nuclear cycle 12–14 in 0–3 hr old, *nos gal4*, $casp^{lof}/Df$, *nos gal4>casp$^{WT}$* and $casp^{lof}/Df$;*nos gal4>casp$^{WT}$* embryos.

in the total number of pH3-positive germ cells simultaneously labeled with the anti-Vasa antibodies. We decided to focus on syncytial blastoderm embryos between NC12-14. As anticipated, maternal overexpression of Casp led to a corresponding increase. In control (*nos gal4*) embryos, roughly 20% PGCs display either 1 or >1 pH3 positive PGCs whereas, 70% of *nos-gal4 >Casp$^{WT}$* embryos show 1 or >1 pH3 positive PGCs (**Figure 6F**). Conversely, less than 10% of *casp$^{lof}$* embryos showed either 1 or >1 pH3 positive PGCs and pole cells with 3 or >3 pH3 positive cells were completely absent in this background as opposed to 25% present in the control embryos (**Figure 6F**). Importantly, maternal expression of the rescue construct in the *casp$^{lof}$* embryos ameliorated the loss of pH3-positive PGCs seen in just the mutant (*casp$^{lof}$*) PGCs.

## Does Casp function affect canonical MBT regulators?

Our analysis thus far has revealed two related yet distinct phenotypes associated with maternal loss of *casp*. First, it can influence proper PGC formation and specification during early embryonic development via its effect on the accumulation of germ cell determinant, Oskar. Subsequently, it helps orchestrate cellular movements leading up to gastrulation during mid-embryogenesis. While the first activity is likely germ cell autonomous, the second relates to its role in the somatic cells/nuclei. Moreover, both the somatic and germline compartments of *casp^lof* embryos share centrosomal as well as cytoskeletal aberrations. We thus wondered whether the two activities are mechanistically connected, and if the possible connection relates to the (MZT). One aspect of the MZT that has been highlighted in recent years is the active degradation of maternal proteins (*Cao et al., 2020*). These proteins comprise 2% of the maternal proteome and are degraded abruptly, at the end of the MZT. In *Drosophila*, ubiquitin-proteasome-based degradation of three repressor proteins, namely Smaug (Smaug), Trailer hitch (Tral), Maternal expression at 31B (Me31B), marks the MZT (*Cao et al., 2020*). We thus decided to probe, using antibodies generated against these proteins (*Figure 7*) if the degradation of any of these proteins is affected due to maternal loss of Casp/*casp* during the initial hours of embryonic development (0–1, 1–2, and 2–3 hr). TER94 protein is unchanged (*Figure 7* D1, D2) as is the α-tubulin loading control (*Figure 7* E1, E2). Furthermore, pattern of degradation of Me31B and TRAL proteins in the *casp^lof* embryos is relatively unaffected, when compared to the *w^1118* control samples.

By contrast, Smaug (*Figure 7* C1, C2) is an interesting exception. In control, 0- to 1-hr-old embryos, low levels of Smaug protein were observed, which increased in the 1- to 2-hr-old embryo extract but decreased the 2–3 hr time-window (*Figure 7C*), data that agrees with previously published work (*Cao et al., 2020*). However, in the *casp^lof/Df* embryos levels of Smaug protein are modestly elevated from the start (0–1 and 1- to 2-hr-old extract samples respectively; *Figure 7* C2). Critically, unlike control embryos, Smaug protein persists in 2- to 3-h r-old *casp^lof/Df* embryonic extracts (*Figure 7* C2, *). Altogether, these data argue that Smaug degradation is specifically adversely influenced in the *casp^lof/Df* embryos at the end of the MZT.

## Germ cell specifc Smaug levels are influenced by casp activity

Since Smaug levels appear to be elevated in embryonic lysates in the MZT, we decided to test if the Smaug levels are elevated specifically in *casp^lof/Df* pole cells. To this end, pre-syncytial as well as early syncytial blastoderm embryos of both the genotypes (control and *casp^lof/Df*) were stained with anti-Smaug and anti-Vasa antibodies. Aligned with the Western blot data presented earlier, levels of Smaug protein are significantly elevated in pre-syncytial embryos (*Figure 8*, compare panel B3 with A3). Also, in *casp^lof/Df* embryos, Smaug protein seems to accumulate in discernible large puncta which are barely visible at this stage in control embryos (*Figure 8*, B3 vs A3). At the syncytial blastoderm stage (Stage 4), however, the Smaug-positive puncta appear larger and more numerous in the few surviving PGCs from the *casp^lof/Df* embryos (*Figure 8* D3) when compared to controls (*Figure 8* C3).

Taken together, our data suggest that Smaug protein levels are appreciably elevated in pole cells from *casp^lof/Df* embryos. The specific increase in Smaug levels may, in part, be due to inappropriate accumulation and/or stabilization of Smaug. Smaug protein is necessary for translational control of the posterior determinant *nanos*. Early reports indicated that unlocalized *nos* RNA is translationally repressed by Smaug which binds to Smaug response elements (SREs) within 3'UTR of *nos* RNA. In addition to *nos*, it also regulates *hsp83* translation. Smaug participates in multiple, overlapping mechanisms including interaction with the components of the translation machinery as well as deadenylation to regulate translation/localization of the target RNAs (*Dahanukar et al., 1999*; *Zaessinger et al., 2006*).

Intriguingly, recent data from Lipshitz lab has, in fact, shown that Smaug protein accumulates in germ granules (*Siddiqui et al., 2023*). Moreover, in addition to its canonical role in regulating *nos* translation, it can also repress *oskar* at the translational level (see Discussion). As the embryonic PGC count depends upon Oskar levels, *Smaug* mutant embryos show an increase in the total number of PGCs. These data fit nicely with our observations and are entirely consistent with a model incorporating the downregulation of Smaug mediated by Casp.

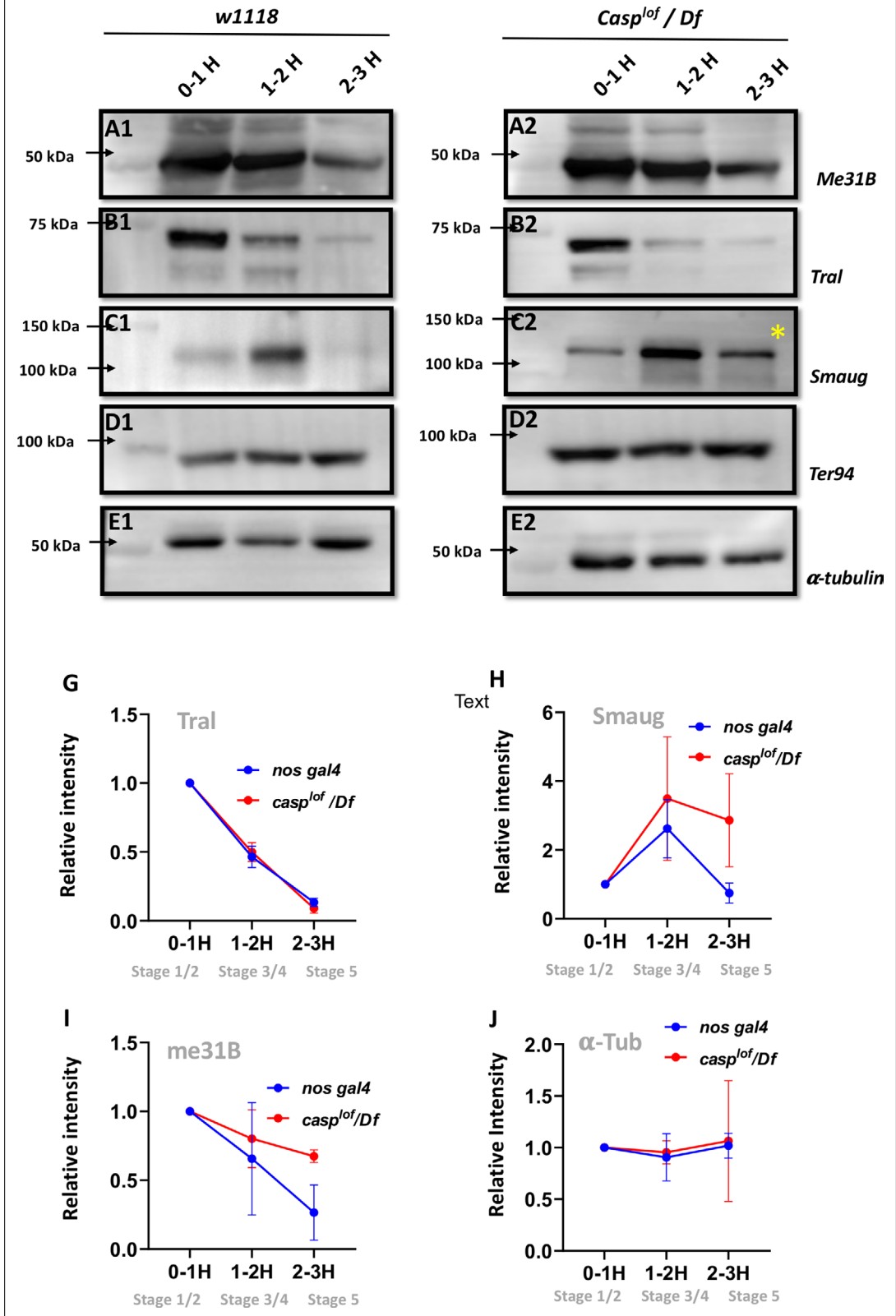

**Figure 7.** Reduction in Casp activity specifically affects Smaug degradation during the MZT. At the MZT, a few maternal proteins are actively degraded (*Cao et al., 2020*). Embryos from mothers with genotype *w1118* (**A1–E1**) and *casp^lof^/Df* (**A2–E2**) were collected using three time intervals (0–1, 1–2, 2–3 hr), and embryonic lysates were separated on SDS-PAGE gels. The western blots were probed with antibodies against ME31 (**A1, A2**), Tral (**B1, B2**) and Smaug (**C1, C2**) to assess the extent of protein degradation, as part of the MZT. α-tubulin (**E1, E2**) was used as a loading control, while TER94

*Figure 7 continued on next page*

*Figure 7 continued*

(**D1, D2**) was used as a negative control as it is a maternal protein that does not undergo degradation at MZT. (**G–J**) Quantitation of the intensity of change of protein band (normalized to 1), for Tral (**G**), Smaug (**H**), ME31B (**I**) and tubulin (**J**). Each data point is an average of over five western blots. The intensities at each time point for *nos gal4* vs *casp^lof^/Df* are statistically not significant.

The online version of this article includes the following source data for figure 7:

**Source data 1.** PDF file containing original western blots for *Figure 9B*, indicating the relevant bands.

**Source data 2.** Original unedited blots for western blot analysis displayed in *Figure 9B*.

## Functional analysis of different protein domains within Casp

To decipher the specific functions of the individual protein domains of Casp during pole cell formation and division we followed an experimental strategy involving rescue of the loss-of-function phenotype. We compared the extent of rescue observed in the presence of individual deletion constructs to the full-length wild-type control. To this end, fly lines expressing domain deletions of Casp under the *pUASp* promoter were generated (*Figure 9A*). The *pUASp-casp^ΔUBA^*, *pUASp-casp^ΔUBX^* and *pUASp-casp^ΔUAS^* construct represent deletions of the UBA, UBX, and UAS domains, respectively. Additionally, we also generated fly lines that express the *pUASp-casp^ΔUASΔUBX^* which deletes both the N and C-terminal domains simultaneously. A full-length coding sequence of Casp was also cloned, and this *pUASp-casp^WT^* line was used as a positive control carrying the 'rescue' construct. A western blot confirmed the expression of these constructs in adult animals with different domain deletions (Δ), *UAS-casp^Δ^* driven by *nos-GAL4* driver (*Figure 9B*). To assess the domain-specific functions of Casp in isolation without the confounding effects of endogenous Casp, the fly lines were balanced with a *casp^lof^* allele on the second chromosome. These balanced lines were crossed to a fly line with the maternal driver *nos-GAL4* on the third chromosome and an allele with a deficiency for *casp (Df)*. Embryos laid by mothers of the genotype *casp^lof^/Df; nos-GAL4/pUASp-casp^Δ^* were used to determine maternal effects of the domain deletions (*Figure 9C*).

Embryos were collected from females of different genotypes and stained using anti-Vasa antibodies. As can be seen by the comparison shown in the bar graph (*Figure 9C*), in terms of the rescue of PGC numbers, deletion of the UBA domain does not affect the rescuing activity of the Casp protein, the UBX domain deletion can rescue, but is weaker than the UBA domain deletion, whereas deletion of the UAS domain is the least effective with regards to the rescue.

To extend these observations, we overexpressed individual Casp deletion mutants in an otherwise wild-type background and counted the total number of PGCs. Neither the maternal overexpression of *pUASp-casp^wt^* nor any of the *pUASp-casp^Δ^* increased lethality in terms of decrease in hatching of embryos, in all cases the hatching was in the range 90–95%. Again, as seen before, overexpression of *casp^wt^* under these conditions led to a significant increase in the total number of PGCs in blastoderm embryos, with average PGC's at 40. The ability to enhance the PGC division is retained in the absence of the UBA domain as, in this particular background, the mean count of the total number of germ cells was similar to embryos that expressed full-length protein. By contrast, deletion of the UAS domain is unable to support the (increased) germ cell division, and the total PGC count dropped to the same level as *nos Gal4* embryos. Interestingly, deletion of UBX domain, yet again, displayed intermediate activity, with the *ΔUBX-ΔUBA* domain deletion also being unable to drive the increase in PGCs.

Proteomic analysis of the embryonic extracts suggested that in addition to TER94 protein, Casp is also associated with VAP (*Figure 9—figure supplement 1A*; *Tendulkar et al., 2022*). We thus sought to test if physical interaction between Casp and VAP is relevant for pole cell formation. To assess this possibility, we used a deletion variant of Casp, Casp^ΔFFAT^, which cannot interact with VAPB (*Tendulkar et al., 2022*). Expression of *casp^ΔFFAT^* under the control of maternal driver, in a *casp^lof^/Df* mother showed hatching (~60%) frequencies at par with wild type (*Figure 9—figure supplement 1B*), with expression levels equal to other *casp* constructs (*Figure 9—figure supplement 1C*). Importantly, *nos-Gal4*-dependent maternal overexpression of *UAS-casp^ΔFFAT^* led to an increase in the total number of PGCs in syncytial blastoderm embryos, comparable to full-length *UAS-casp^wt^* (average PGC ~23). Lastly, expression of *casp^ΔFFAT^* could rescue the PGC loss upon maternal loss of *casp* (compare *Figure 9—figure supplement 1* F1-3 to *Figure 9C*). These data suggested that interaction between Casp and VAP is either not essential or partially redundant for Casp function in the context of determining PGC.

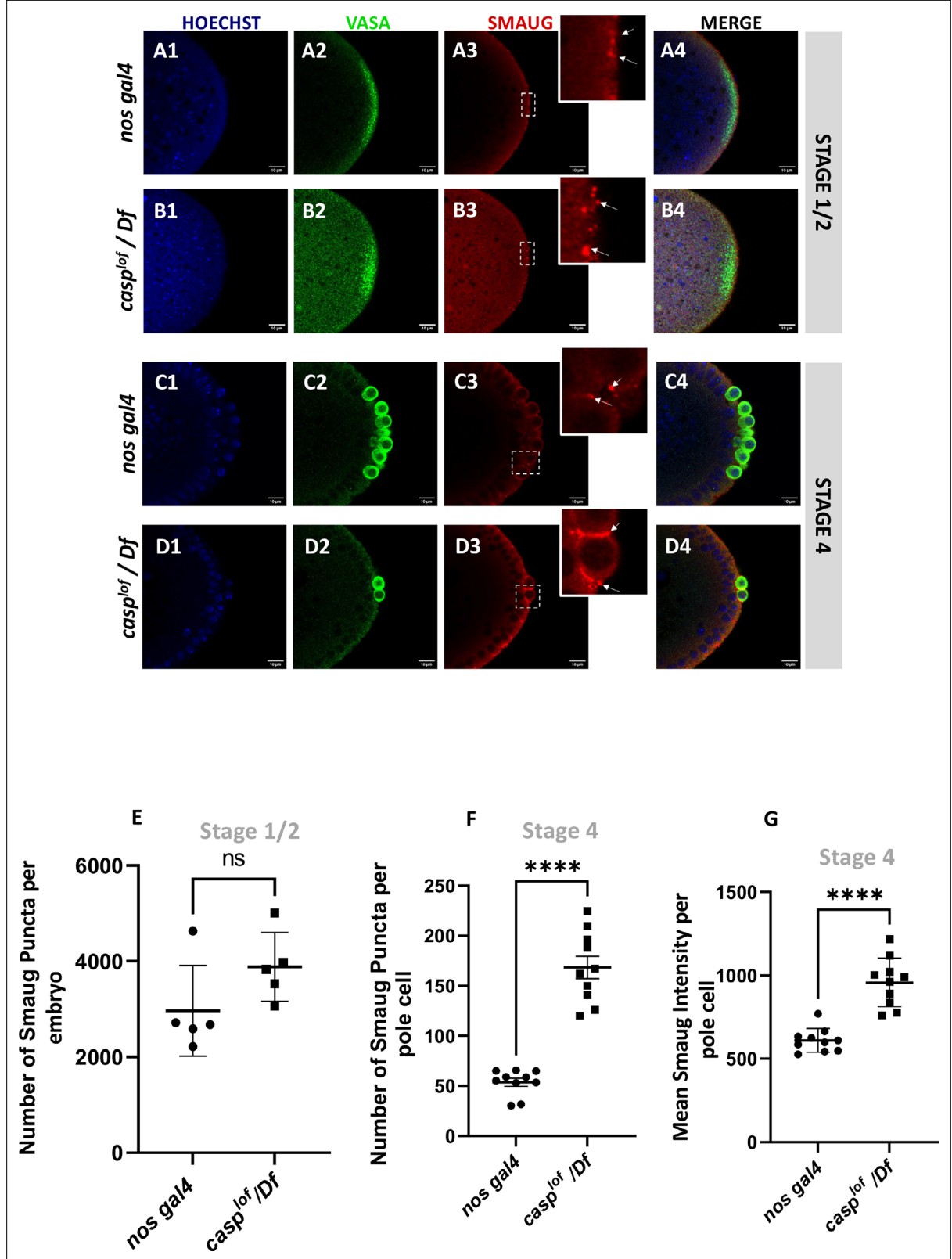

**Figure 8.** Decrease in Casp correlates with an increase in Smaug levels in the pole buds and PGCs. Confocal microscopy images of the posterior termini of stage 1–2, and 4 embryos derived from *nos gal4* and *casp*[lof]/*Df* females. Embryos were immunostained with Vasa (green) and Smaug (red) antibodies. Hoechst marks the nuclei (**A–D**). Smaug expression was quantified across the two genotypes and plotted as bar graphs (**E–G**). N=5 embryos (Stage 2) and 10 embryos (Stage 4).

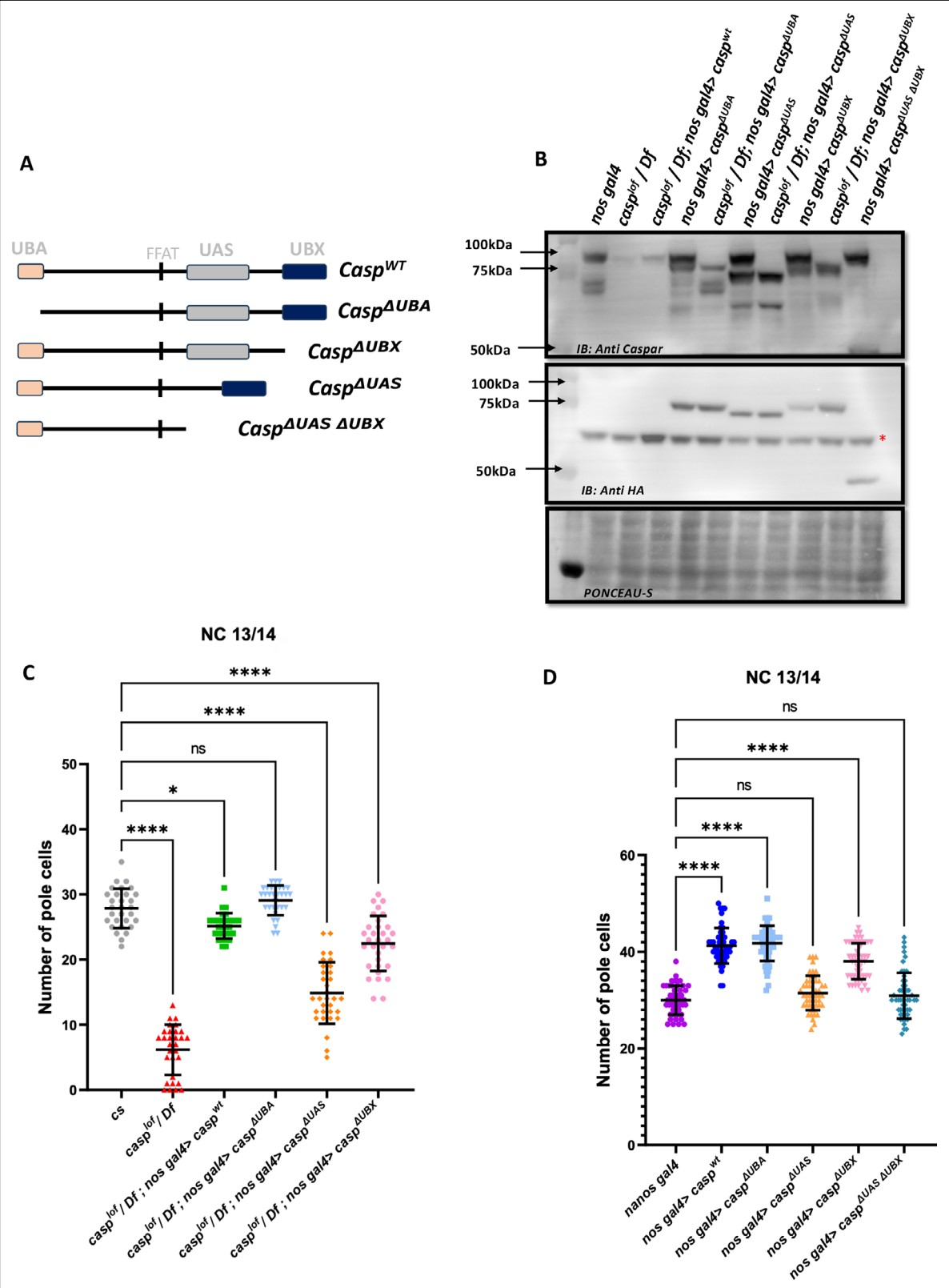

**Figure 9.** Structure-function analysis of Casp protein domains in regulating pole cell number. (**A**) Schematic representation of different domain deletion variants of Casp. As shown in the schematic, the WT Casp protein consists of different functional domains including UBA, UAS, UBX and the FFAT-like motif. (**B**) Western blot of the truncated Casp proteins, deficient in the domains indicated, probed with rabbit anti-casp (1:10,000) and rabbit anti-HA (1:2000) antibody. *nos gal4* served as a control. Ponceau staining was used as a loading control. Asterisk (*) denotes non-specific antibody binding.

*Figure 9 continued on next page*

*Figure 9 continued*

(**C**) The total number of germ cells from nc 13/14 embryos derived from mothers expressing different *casp* domain deletion constructs in the *casp^lof^*/ *Df* background were marked with Vasa antibodies, quantified and plotted as a bar graph. n=30, Ordinary one-way ANOVA, (****) p<0.0001, (*) *p<0.05*. (**D**) The total number of germ cells from nc 13/14, derived from mothers expressing casp domain deletion, in an otherwise *wild-type* background, were stained with anti-Vasa antibodies. Vasa positive cells (i.e. PGCs) were quantified and plotted as a bar graph. n=30, Ordinary one-way ANOVA, (****) p<0.0001.

The online version of this article includes the following source data and figure supplement(s) for figure 9:

**Source data 1.** PDF file containing original western blots for *Figure 9B*, indicating the relevant bands.

**Source data 2.** Original unedited blots for western blot analysis displayed in *Figure 9B*.

**Figure supplement 1.** Casp:VAP interaction is not required for defining pole cell number.

**Figure supplement 1—source data 1.** PDF file containing original western blots for *Figure 9—figure supplement 1* indicating the relevant bands.

**Figure supplement 1—source data 2.** Original unedited blots for western blot analysis displayed in *Figure 9—figure supplement 1*.

Altogether, the functional analysis of different protein domains within Casp underscores the importance of UAS and UBX domains, especially in embryonic contexts, including PGC development.

## Discussion

Proper development of biological systems in an organismal context involves complex interactions between various individual pathways. The qualitative nature of the interaction(s) between pathway components ultimately determines how different pathways intersect in a context-specific manner. Curiously however, there are only a defined number of signaling pathways/circuits that have been elucidated thus far. Consistently, the entire molecular cassettes that constitute a given pathway or a select number of pathway components, are reiteratively used in a variety of biological contexts even within a lifecycle of the same organism. Moreover, while such repurposing is relatively frequent, the corresponding biological outcomes are diverse in nature and insightful in ways more than one. Our data detailing the functional involvement of Casp and TER94 in the PGCs is a case in point. Casp function was initially characterized in the context of *Drosophila* immune response (*Kim et al., 2006*). However, the original study did not investigate the developmental roles of Casp and its interactors. Here, we report novel activities of *Drosophila melanogaster* immune components Casp and TER94 during early embryonic development with a specific focus on germline.

Early embryonic patterning is a dynamic process in metazoans which is primarily regulated by the deposition of maternal gene products including RNAs and proteins. Such maternal determinants are localized in a spatially restricted manner (*Schier, 2007*). On many instances, the unique pattern of localization of specific factors underlies their activities during embryo patterning. To execute their functions properly, patterning determinants heavily rely upon the members of the housekeeping machinery that carefully calibrate the synthesis, stability, transport, and degradation of diverse regulatory components. Here, we have explored novel embryonic functions for the FAF1 ortholog, Casp in *Drosophila*. Initially, we uncovered that maternal loss of *casp* activity results in partially penetrant embryonic lethality. The *casp* allele was determined to be a strong loss of function, but not a null. The partial penetrance could also be due to redundancy. Consistent with either possibility, quantitation of the individual phenotypes yielded significant but variable penetrance.

Our data demonstrate Casp's involvement during cellular movements that lead to gastrulation, a likely cause underlying the lethality. Surprisingly, Casp protein is enriched in the PGCs, which are specified at the posterior pole under the control of the master germ cell determinant, Oskar. PGC formation and specification in a young *Drosophila* embryo depends on the posteriorly anchored specialized cytoplasm (or pole plasm) enriched in RNA and protein components essential to determine germ cell identity and behavior. Two important traits distinguishing early pole cells from the surrounding somatic nuclei include precocious budding and limited mitotic self-renewal. Consistent with its functional involvement in both these processes, maternal loss of either Casp or its protein partner TER94 resulted in a reduced number of buds and subsequent loss of PGCs. While qualitatively similar, the severity of phenotypic consequence due to loss of Casp and TER94 regarding PGC loss, is not identical. Interestingly, however, both proteins appear to influence centrosome behavior and dynamics that are of paramount significance in forming pole cells in *Drosophila* embryos.

Thus far, Germ cell-less (Gcl) is the only protein shown to control PGC formation in a similar manner (*Jongens et al., 1992*; *Cinalli and Lehmann, 2013*). Importantly Gcl activity depends on its ability to influence proper separation of centrosomes and elaboration of astral microtubules in dividing pole buds (*Lerit et al., 2017*). Aberrant centrosome behavior in *gcl* mutant embryos adversely affects PGC budding and equitable distribution of germ plasm among daughter cells. Intriguingly, these phenotypic traits can be recapitulated by simply engineering defective centrosome separation (*Lerit et al., 2017*). It is thus noteworthy that maternal loss of Casp and TER94 leads to similar defects and future experiments will reveal mechanistic details underlying roles of these two proteins in regulating centrosome dynamics in PGCs and their possible interaction with Gcl and its protein partners. Of note, a germplasm interactome (*Thomson et al., 2008*) included TER94 along with other important germ plasm proteins including Vasa, Tudor, and others.

TER94, an ER protein, is a major Casp interactor. Our data also indicate that in addition to PGC formation, TER94 and Casp regulate early nuclear division cycles in syncytial blastoderm embryos and subsequent cell divisions in the somatic compartment cell cycle processes in the gastrulating embryos. (Casp and TER94 are distinct compared to Gcl in this regard which has a strictly germ-cell-specific function). Taken together, these observations suggest that Casp and TER94 contribute to critical early developmental functions leading to mid-blastula transition which precedes gastrulation and germband extension. In the mammalian context, the FAF1-VCP interaction is mediated by the UBX domain. Thus, it is unsurprising that deletion of the protein domain crucial for association between Casp and TER94 resulted in a somewhat diminished function as compared to a full-length version. Curiously however, deleting UAS fragment, a protein domain of unknown function, implicated in self-association, resulted in substantially compromised activity as compared to native protein. Future experiments will focus on the specific molecular interactions this (and other) individual domain(s) are involved in. It will be also important to determine how protein degradation especially engineered via ubiquitin modification contributes to Casp stability and function.

Intriguingly, our data also argue that regulation mediated by *casp* is likely critical for maintenance of PGC fate via maintenance of Oskar levels. Importantly, several independent observations suggest that this influence is likely zygotic. First, barely detectable amount of Casp protein is deposited in the egg and bulk of the Casp protein is generated by translation post-fertilization. Second, Casp protein controls overall levels of Smaug at the post-transcriptional level including in the early embryonic PGCs. It was recently reported that Smaug protein accumulates in the germ granules where it controls Oskar (and Bruno) translation negatively by binding to Smaug response elements present in the untranslated regions within the respective RNAs (*Siddiqui et al., 2023*). Consistently loss of function mutations in *Smaug* result in a modestly elevated PGC count in stage 4–5 embryos whereas a reciprocal phenotype is observed in *casp^{lof}* embryos. Taken together with the increase in Smaug levels in *casp^{lof}* embryos, it would be reasonable to propose that PGC-specific phenotypes observed upon loss of *casp* are, in part, mediated by excess accumulation of Smaug. Moreover, these authors also suggest that Smaug dependent regulation of *oskar* RNA likely has a significant zygotic i.e. embryonic component which aligns well with our data. Future experiments will be necessary to elucidate the mechanism underlying regulation of Smaug levels by Casp. It will also be interesting to examine if Smaug and Casp regulate centrosome behavior reciprocally to control the final PGC count.

Degradation of maternally deposited RNAs and proteins constitutes an important transition during embryonic development. Zygotic Genome Activation (ZGA) and turning over of maternal determinants (both RNAs and proteins) happen almost simultaneously. Together these two events constitute MZT which is delayed in the germ cell compartment as opposed to soma. Nonetheless, recent data have suggested that the two events possibly occur in a coordinated manner possibly via shared components (*Colonnetta et al., 2023*). Many of the embryonic phenotypes that *casp^{lof}* embryos display (aberrant nuclear migration, cellularization defects, defective gastrulation etc.), are shared by mutations in gene products either directly or indirectly involved in one of these processes. It will be important to elucidate new interactors of Casp as its function likely impacts protein degradation and/or stability of the target proteins. Recent reports have indicated potential involvement of ZGA regulators such as Zelda and CLAMP in germline/soma distinction. Critically this function of the components of ZGA depends on proper anchoring, release and transmission of posteriorly anchored pole plasm that involves centrosome function. Maternally compromising ZGA components resulted in

inappropriate release and transmission of pole plasm RNAs, a phenotype that is partially recapitulated in *casp^lof* and *TER94i* embryos.

While our data have clearly established the involvement of Casp and TER94 during PGC formation and specification, further studies will be necessary to elucidate their precise molecular function(s) underlying this activity. Several observations are noteworthy in this regard and will guide the course of future investigation. The first set of results points to the possible participation of both Casp and TER94 during ubiquitin-dependent protein degradation.

A regulatory network incorporating Bru1, Cup, Oskar, and Smaug are key to PGC specification. The Skp Cullin and F-box (SCF) containing complex marks proteins such as Smaug for degradation by functioning as an E3 ubiquitin ligase (*Cao et al., 2020*; *Cao et al., 2022*; *Figure 10A*). The ubiquitinated Smaug is then degraded by the proteasome. In the *mammalian* context, the Casp/TER94 orthologs FAF1/VCP (in *green* font; *Figure 10A*) have been found to interact with the SCF complex. Our data supports a similar scenario in *Drosophila*.

It would be reasonable to propose that Casp recruits TER94 via Casp's UBX domain (*Figure 10B*). Independent proteomic studies (*Thomson et al., 2008*; *DeHaan et al., 2017*; *Tendulkar et al., 2022*; *Figure 10B*) suggest multiple overlaps between Casp/TER94 and germ-cell-specific protein complexes (*Figure 10B*), again suggesting functional relationships between Casp/TER94 and germ cell determinants. In addition, the SCF complex contributes to cell cycle progression and integrity of centrosomes (*Wojcik et al., 2000*; *Murphy, 2003*; *Phuong Thao et al., 2006*; *Figure 10C*). Total PGC count in early embryos depends on centrosome function and efficient mitotic divisions. So SCF complex based degradation may have multiple targets that participate during germ cell formation and specification (*Figure 10C*).

Casp likely functions as an adapter protein that works in conjunction with the SCF complex (*Cao et al., 2020*; *Cao et al., 2022*; *Figure 10D*) and TER94 (*Figure 10D*). The binary complex between TER94 and Casp subsequently targets ubiquitinated proteins such as Smaug which are present in a complexed form with other proteins. Homeostatic maintenance of Smaug levels via Casp-mediated proteolytic degradation could explain many of our observations. Consistently, increased Smaug levels, due to diminished Casp function leads to loss of PGCs, whereas *smaug* mutant embryos show the opposite phenotype (*Siddiqui et al., 2023*). Intriguingly, even though Smaug and Oskar are members of a complex, the reduction in Casp activity results in the stabilization of Smaug. By contrast, Oskar levels are diminished resulting in loss of PGCs. The mechanism underlying the substrate-dependent divergent activities is not known at this point.

The similarity between the phenotypic consequences due to loss of ZGA components and Casp/TER94 also needs a special mention. The only study implicating TER94 during early embryogenesis (*Zeng et al., 2014*) suggested that TER94 can potentiate BMP signaling. Interestingly, *decapentaplegic* (*dpp*), a BMP ligand which is one of the important targets of ZGA regulators including *Zelda*, can also influence the specification of embryonic PGCs (*Colonnetta et al., 2023*). PGCs need BMP signals (Dpp) to maintain their identity. Furthermore, the exposure to BMP signals needs careful calibration as excess BMP signaling in the surrounding soma leads to PGC loss. Importantly, components of protein degradation machinery including those involved in ubiquitination (Smurf, a Ubiquitin ligase) and sumoylation (Ubc9) appear to be involved in fine-tuning the signaling (*Deshpande et al., 2014*). Thus, it will be of interest to determine if the aberrant germ cell specification observed due to loss of *casp* and the components of BMP signaling pathway is mechanistically connected. This seems especially relevant in the light of their respective dependence on the components of the protein degradation machinery. It will be of considerable interest to investigate whether and how, zygotic activity of Smaug, one of important canonical MBT regulators, fits into this picture. In sum, while the specific details await detailed examination, it is apparent that possible recapitulation of maternal regulation in the zygotic context may be a recurrent theme rather than an isolated anomaly.

## Materials and methods
### Fly husbandry and stocks

Flies were raised on standard cornmeal agar at 25 °C unless stated otherwise. *Casp^lof* (11373), *Casp Df* (23691), *nos Gal4* (4937), *Mat α4-tubulinGal4:VP16* (7063) and *TER94 RNAi* (32869) lines were

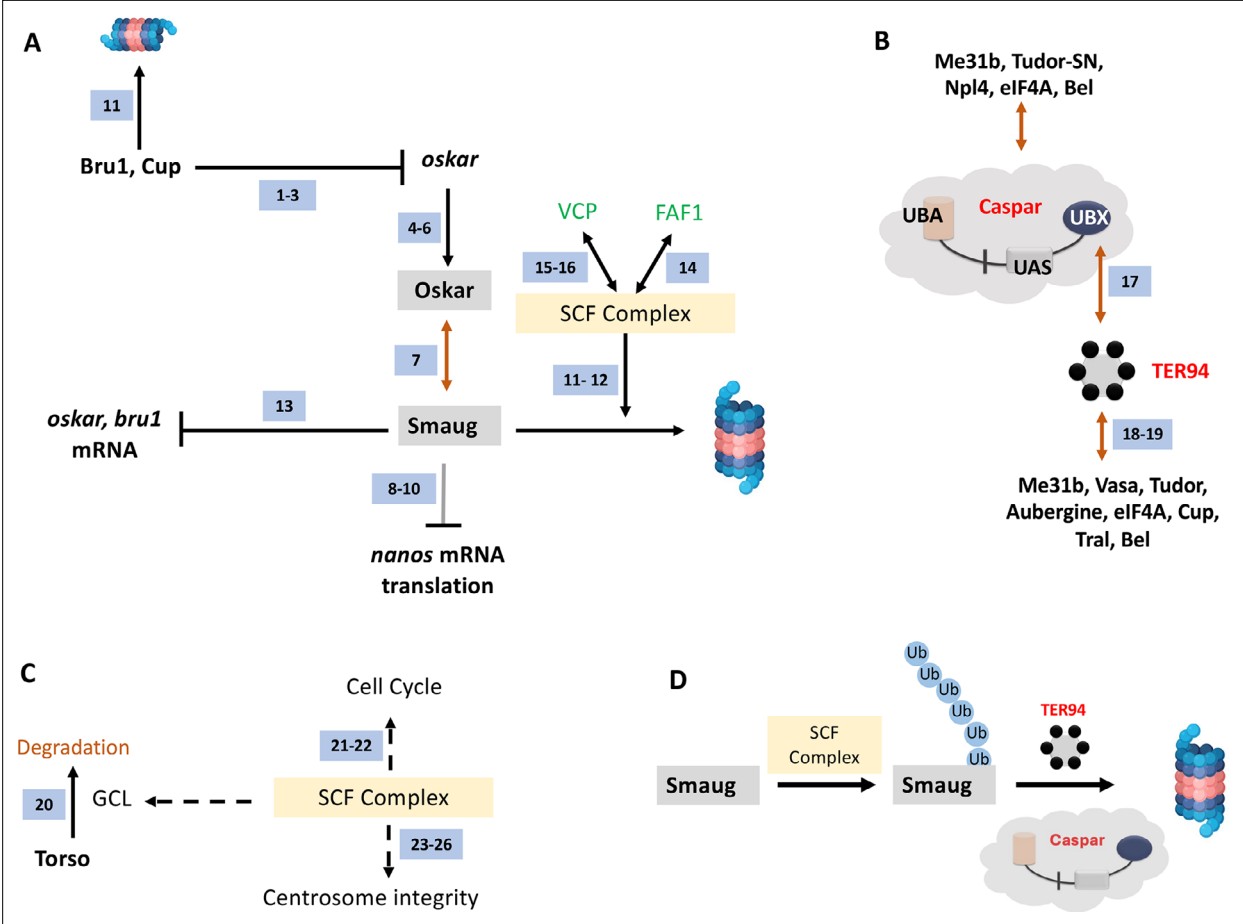

**Figure 10.** A model for Casp function in PGCs. (**A**) Smaug is marked for degradation by the SCF complex. Specific pole plasm components including Cup, Oskar, and Smaug are actively degraded. *oskar* is regulated by Bru1 and Cup (*ref 1–3*; *Nakamura et al., 2004*; *Kim et al., 2015*; *Bayer et al., 2023*). Oskar (*ref 4–6*; *Mahowald, 2001*; *Huynh and St Johnston, 2004*; *Lehmann, 2016*) is the master determinant of PGC fate, and hence the stability of the Oskar:Smaug complex (*ref 7*; *Kubíková et al., 2023*) is a key to PGC determination and proper specification. Oskar also regulates posterior cell fate by regulating *nos* translation which is modulated by Smaug (*ref 8–10*; *Dahanukar et al., 1999*; *Zaessinger et al., 2006*; *Jeske et al., 2011*). The SCF complex is a multi-protein E3 ubiquitin ligase complex, and Smaug is one of its prominent targets (*ref 11–12*; *Cao et al., 2020*; *Cao et al., 2022*). Smaug appears to repress the translation of *oskar* and *Bruno 1* mRNA (*ref 13*; *Siddiqui et al., 2023*). In mammals (marked with green font), FAF1 modulates the SCF complex (*ref 14*; *Morais-de-Sá et al., 2013*) with VCP/p97 assisting in the degradation of ubiquitinated Smaug (*ref 15–16*; *Li et al., 2014*; *Reim et al., 2014*), suggesting potential conservation of the activities in the fly orthologs, Casp and TER94. (**B**) Casp/TER94 interact with an overlapping set of proteins. TER94 has earlier been demonstrated to be associated with Oskar (*Ruden et al., 2000*). Our data points to an association between *Drosophila* Casp and TER94 (this study and *ref 17*; *Tendulkar et al., 2022*) that participates in protein degradation. In a proteomic analysis of germ cell components, TER94 was identified (this study and *ref 18*; *Thomson et al., 2008*), along with other bona-fide germ cell constituents including Cup, Tral, Bel, eIF4A, Tud and Vasa (*ref 18* and *19 Thomson et al., 2008*; *DeHaan et al., 2017*). Both, Casp and TER94 are thus enriched in the pole cells and interact with proteins that specify PGC fate. (**C**). The SCF Complex, Gcl and Centrosome integrity. The SCF complex is localized to the centrosome. Gcl interacts with Cullin 3 to degrade Torso receptor to promote PGC fate (*ref 20*; *Pae et al., 2017*). The SCF complex regulates the cell cycle (*ref 21–22*; *Margottin-Goguet et al., 2003*; *Rogers et al., 2009*) and is a known regulator of centrosomal integrity (*ref 23–26*; *Wojcik et al., 2000*; *Murphy, 2003*; *Phuong Thao et al., 2006*; *Cunha-Ferreira et al., 2009*), thereby influencing PGC specification (*Lerit et al., 2017*). (**D**) Casp and TER94 assist in the degradation of Smaug. The SCF complex works as a ubiquitin E3 ligase to mark Smaug for Degradation and PolyUb-Smaug is degraded by the action of Casp and TER94 during the late-MZT.

procured from the Bloomington *Drosophila* Stock Centre, with numbers in brackets indicated Stock numbers.

## Cloning of *casp* deletion constructs and generation of transgenics

UASp-casp$^\Delta$ constructs (*pUASp casp$^{\Delta UBA}$, pUASp casp$^{\Delta UAS}$, pUASp casp$^{\Delta UBX}$, pUASp casp$^{\Delta UAS\Delta UBX}$*) were generated, starting from a *pUASp-attB-Casp* construct (*Tendulkar et al., 2022*), with the design including an N-terminal HA tag. The constructs were injected into a *w$^{1118}$*; *Attp2* embryo in the

NCBS-CAMP transgenic injection facility, and stable lines were generated by balancing against a *w^1118*; *TM3Sb/TM6Tb* animal.

For cloning and amplification, a functional N-terminal HA tag was introduced with the 5'-forward primers and a 3'-homology arm was introduced using the 3'-reverse primers. The exception was the *pUASp Casp^ΔUAS* construct where two fragments, one upstream and one downstream of the UAS domain were amplified separately with 22 nucleotide homologous overhang between them to mediate homologous recombination. The upstream fragment contained the HA sequence while the downstream fragment contained the 3'-homology arm. The sequences of the primers used are as follows. *TGTTCCAGATTACGCTGGCGGC* was used as the 5'-forward primer for the *pUASp casp^wt*, *pUASp casp^ΔUBX*, *pUASp casp^ΔUASΔUBX* and the upstream fragment of the *pUASp casp^ΔUAS* constructs. *TGTTCCAGATTACGCTGGCGGCCCCCATCCTATCCTGGTGCC* was used as the 5'-forward primer for the *pUASp casp^ΔUBA* construct. *CGGATGATGAGATAAGTGGCTCCACGGAAACATGCGAAATGTTTGAGGAGCAG* was used as forward primer for amplifying downstream fragment of the *pUASp casp^ΔUAS* construct. *ACCATGGGTTTAGGTATAATGTTATCAAGCTCC* was the reverse primer for *pUASp casp^wt*, *pUASp casp^ΔUBA*, and downstream fragment of *pUASp casp^ΔUAS*. *TTAGGTATAATGTTATCAAGCTCCTCATTCGGACGGCTCCTGAGGTAG* was the reverse primer for *pUASp casp^ΔUBX*. *TTAGGTATAATGTTATCAAGCTCCTCACGTGGAGCCACTTATCTCATCATCCG* was used as the reverse primer for *pUASp casp^ΔUASΔUBX*. *CGTGGAGCCACTTATCTCATCATCCG* was the reverse primer for upstream fragment of *pUASp casp^ΔUAS*. PCR products were further amplified with a common 5'-primer *ATAGGCCACTAGTGGATCTGATGTACCCATACGATGTTCCAGATTACGCTGGCGGC* and their respective reverse primers to introduce the 5'-homology arm. Inserts were recombined into *pUASp AttB* vector linearised at the BamH1 site, using a variation of the SLiCE cloning method (*Zhang et al., 2012*; *Zhang et al., 2014*). In short, *E. coli* DH10B expressing the optimised $\lambda$-prophage red recombinase system (PPY cells) was cultured in the presence of arabinose to induce recombinase expression and subsequently used for preparing competent cells. DNA fragments containing homologous overhangs were co-transformed into the competent bacteria and resultant colonies were screened through PCR to identify proper recombinants and subsequently confirmed through sequencing. The pUASp *casp^ΔFFAT* was PCR amplified from *pRM-HA:casp ^ΔFFAT* (*Tendulkar et al., 2022*), cloned into pUASp-AttB, sequenced for validation and injected into a *w^1118*; *Attp2* animal.

## Embryonic lethality

0–3 hr embryos were collected, transferred to a fresh sugar-agar plate, and unhatched larvae were scored after 48 hr to determine viability.

## Immunoprecipitation

0–3 hr embryos were lysed in Co-IP Lysis Buffer (20 mM Tris pH 8.0, 137 mM NaCl, 1% IGEPAL, 2 mM EDTA, 1 X PIC) using a Dounce homogenizer, and centrifuged at $21,000 \times g$ for 30 min. 3 mg of total lysate was incubated with 5 μg of primary antibody (Rb anti-Casp) and 5 μg of Normal Rabbit IgG overnight at 4 °C. Antigen-antibody complexes were captured using 50 μL of Bio-Rad SureBeads Protein A (1614013) at 4 °C for 4 hr. Beads were washed six times with Co-IP Lysis Buffer and protein complexes eluted by boiling in 1 X Laemmli Sample Buffer. Eluted proteins were resolved on a 10% polyacrylamide gel followed by western blotting or in-gel trypsin digestion, described in the following sections.

## Western blot analysis

Embryos collected at varied time points (0–3, 0–1, 1–2, 2–3 hr) were lysed in RIPA buffer (50 mM Tris-Cl, 150 mM NaCl, 0.1% SDS, 0.01% Sodium azide, 0.5% sodium deoxycholate, 1 mM EDTA, 1% Triton X-100, 1 X PIC) with a pellet pestle (Kontes). Lysates were cleared by centrifugation at $21,000 \times g$ at 4 °C for 30 minutes. Protein concentration was estimated using a BCA assay (Pierce) and 30–40 μg of total protein was loaded onto the gel after boiling in 1 X Laemmli Sample Buffer. Proteins separated by 10% SDS-PAGE were transferred onto a PVDF membrane (Immobilon-E, Merck) and blocked in 5% milk in Tris-Buffer Saline (TBS) with 0.1% Tween 20 (TBS-T) for an hour. Blots were then incubated overnight with primary antibody diluted in 5% milk in TBS-T, at 4 °C. Following three washes with TBS-T, blots were incubated with secondary antibodies diluted in 5% milk in TBS-T, for 1 hour at room temperature. Blots were washed thrice with TBS-T and visualized on a LAS4000 Fuji

imaging system after incubating with Immobilon Western Chemiluminescent HRP substrate (Merck). The following antibodies were used: Rabbit anti-VAP, 1:10000 (*Tendulkar et al., 2022*), Mouse anti-α-Tubulin, 1:10000 (T6074, Sigma), Mouse anti-Ubiquitin, 1:1000 (P4D1, Santa Cruz Biotechnology), Mouse anti-HA, 1:5000 (H3663 HA-7, Sigma-Aldrich), Rabbit anti-HA, 1:2000 (04–902 DW-2, Sigma-Aldrich), Rabbit anti-Casp, 1:10000 (*Tendulkar et al., 2022*), 1:2000 (04–902 DW-2, Sigma-Aldrich). Goat anti-rabbit HRP, Goat anti-rat HRP and Goat anti-mouse HRP secondary antibodies, each at 1:10000 (Jackson ImmunoResearch). Rabbit anti-me31B (1:1000), Rabbit anti-Smaug (1:500), Rat anti-Tral (1:1000) were a kind gift from Elmar Wahle (*Cao et al., 2020*).

## In-gel trypsin digestion and LC-MS/MS analysis

Before in-gel trypsin digestion of the Co-IP eluate, the antibody was crosslinked to the SureBeads using DMP (Sigma) according to the NEB crosslinking protocol to avoid elution of the antibody. After crosslinking 10 µg Casp antibody, Co-IP was performed as described above. In-gel trypsin digestion was carried out as previously described (*Shevchenko et al., 2006*). Briefly, Coomassie-stained bands on the gel were excised and cut into 1 mm cubes. Gel pieces were transferred to a clean microcentrifuge tube and destained with buffer containing 50% acetonitrile in 50 mM Ammonium bicarbonate. Reduction and alkylation were carried out on the destained gel pieces by incubating with 10 mM dithiothreitol (DTT) followed by incubating with 20 mM iodoacetamide. Gel pieces were saturated with sequencing grade Trypsin (Promega) at a concentration of 10 ng/µl and incubated overnight at 37 °C. Peptides were extracted by sequential addition of 100 µl of 0.4% Trifluoroacetic acid (TFA) in 10% ACN, 100 µl of 0.4% TFA in 60% ACN and 100 µl of ACN. The pooled extract was dried in a vacuum centrifuge and reconstituted with 50 µl of 0.1% TFA. The peptides in TFA were purified using the StageTip protocol (*Rappsilber et al., 2007*). LC–MS/MS analysis was performed on the Sciex TripleTOF6600 mass spectrometer interfaced with an Eksigent nano-LC 425. Tryptic peptides (1 µg) were loaded onto an Eksigent C18 trap (5 µg capacity) and subsequently eluted with a linear acetonitrile gradient on an Eksigent C18 analytical column (15 cm ×75 µm internal diameter). A typical LC run lasted 2 hr post loading onto the trap at a constant flow rate of 300 nl/min with solvent A consisting of water + 0.1% formic acid and solvent B consisting of acetonitrile. The gradient schedule for the LC run was 5% (vol/vol) B for 10 min, a linear gradient of B from 0% to 80% (vol/vol) over 80 min, 80% (vol/vol) B for 15 min and equilibration with 5% (vol/vol) B for 15 min. Data was acquired in an information-dependent acquisition (IDA) mode over a mass range of 300–2000 m/z. Each full MS survey scan was followed by MS/MS of the 15 most intense peptides. Dynamic exclusion was enabled for all experiments (repeat count 1; exclusion duration 6 s). Peptides were identified and quantified using the SCIEX ProteinPilot software at a false discovery rate (FDR) of 1%. A RefSeq *Drosophila* protein database (release 6) was used for peptide identification. Proteins that were identified in two or more replicates were tabulated.

## Fixation, immunostaining, and imaging of embryos

0–3 hr embryos were collected in a sieve and dechorionated in 4% sodium hypochlorite for 90 s. After thorough washes with distilled water, embryos were fixed in a 1:1 heptane:4% PFA solution for 20 min. The PFA layer was removed, and embryos in heptane were re-constituted with an equal volume of methanol. Embryos were devitellinized by vigorous shaking in the 1:1 heptane:methanol mixture. The heptane layer and the interphase containing non-devitellinized embryos were carefully removed. Devitellinized embryos in the bottom methanol phase were washed twice with methanol and stored at –20 °C till they were ready to be imunostained. For phalloidin and Smaug staining, embryos fixed in heptane: 4% PFA were hand de-vitellinized, after which the standard immunostaining procedure was followed. For immunostaining, embryos were rehydrated by washing thrice with 0.3% PBS-TritonX 100 (PBS-T) for 15 min each. Embryos were blocked in 2% BSA in 0.3% PBS-T for 1 hr at room temperature (RT). Embryos were incubated at 4 °C overnight with primary antibodies diluted in 2% BSA in 0.3% PBS-T at the appropriate dilutions. Following three 15-min washes with 0.3% PBS-T, embryos were incubated in the appropriate secondary antibodies for 1 hr at RT. The following antibodies were used: Rabbit anti-Casp, 1:1000 (*Tendulkar et al., 2022*); Rat anti-α-vasa; 1:50 (DSHB); Rabbit anti-VCP, 1:200 (#2648, Cell Signaling Technology); Rabbit anti-Oskar, 1:1000 (generated by Mandy Jeske, Anne Ephrussi lab); Rabbit anti-Smaug, 1:50; Mouse anti-γ-tubulin, 1:1000 (T6557, Sigma), Rabbit P-Histone H3 (S10) 1:300 (#9701, Cell Signalling); Alexa Fluor 568 Phalloidin, 1:1000

(Invitrogen) The following secondary antibodies were used, goat anti-mouse Alexa488 /goat anti-rabbit Alexa568 /goat anti-rat Alexa647 /goat anti-rabbit Alexa647 /goat anti-rabbit Alexa480, 1:1000 (DSHB). Embryos were washed thrice with 0.3% PBS-T, and DAPI/Hoechst (1:500) was added in the penultimate wash. Embryos were mounted in 70% glycerol and observed under a Leica sp8 confocal microscope with a 20 X objective or Zeiss anisotropy confocal microscope under 63 x objective.

### Live imaging of embryos

Embryos at the appropriate stage were washed and dechorionated as described previously. A 2-well Nunc Lab-Tek II Chamber Slide System (Thermo Fisher Scientific) was affixed with a 3 M Scotch double-sided tape, and dechorionated embryos with intact vitelline membranes were mounted on the tape under a dissecting microscope. Halocarbon oil 200 was used to cover the embryos to prevent dehydration during imaging. Time-lapse bright field imaging of embryos was performed on an inverted LSM confocal system (Zeiss multiphoton 710) at 20 X for ~8 hr. Embryo images were acquired at 5 min intervals.

### Quantification of western blots

Western blots were quantified using ImageJ. Each protein was normalized to the tubulin loading control, and the highest signal for each was set to 1 and all other tubulin band intensities were normalized against it. For MZT experiments, western blots were used to determine protein degradation dynamics. 3–5 biological replicates, representing 3–5 independent embryo collection were assayed. Band intensities were quantified using ImageJ and normalized to α-tubulin as loading control. For each replicate intensities were normalized to the first time point (0–1 hr). Significance thresholds are presented in the figures and figure captions.

### Quantification of the Smaug immunostaining

All quantification of Smaug stained images were done on ImageJ. ROIs were made for a single pole cell through the z-stacks, thereby only analysing one pole cell at a time. For each embryo five such pole cells were measured. To quantify intensity, mean intensity was measured for each ROI per stack and added to get the final intensity. Each data point on the graph represents an average of all five cell intensities per embryo. For Smaug punctae measurement, a combined ROI for each cell was taken and duplicated. Each of the stack was then analysed using 3D objects counter, where thresholding for all images was kept within the range of 80–90 with a minimum size filter kept at zero. The number of objects counted was taken as the total number of Smaug punctae per cell. Five such pole cells were analysed per embryo and the average was plotted as a single data point on the graph. For quantification of poleplasm volume in the embryos, embryos were stained with Oskar. An ROI (kept same for all images) encompassing the posterior part of the embryo was drawn. Gaussian blur 3D (X sigma = 2, Y sigma = 2 and Z sigma = 2) was applied to each image. The volume of pole plasm was analysed by using 3D objects counter after appropriate thresholding.

## Acknowledgements

Stocks obtained from the Bloomington *Drosophila* Stock Center (NIH P40OD018537) were used in this study; Elmar Wahle and Christiane Rammelt for the kind gift of anti-Tral, anti-Me31B, and anti-Smaug antibodies; Snehal Patil and Yashwant Pawar for fly media and stock maintenance; IISER Microscopy facility, Dr Santosh Podder, and Vijay Vittal, for training and maintenance. Science & Engineering Research Board (SERB) grant CRG/2018/001218 to GR. Pratiksha Trust Extra-Mural Support for Transformational Aging Brain Research grant EMSTAR/2023/SL03 to GR, facilitated by the Centre for Brain Research (CBR), Indian Institute of Science, Bangalore. GD's visits to IISER Pune (2023–2025) are supported by the Ministry of Education (MoE) Scheme for the Promotion of Academic & Research Collaboration (SPARC), Grant ID SPARC-1587, managed by IIT Kharagpur, which facilitates collaboration between IISER Pune and Princeton University. The IISER *Drosophila* media and Stock centre are supported by the National Facility for Gene Function in Health and Disease (NFGFHD) at IISER Pune, which in turn is supported by an infrastructure grant from the DBT, Govt. of India (BT/INF/22/SP17358/2016). JS, NW and AR are undergraduates supported by INSPIRE/KVPY fellowships. SD and SH are graduate students supported by research fellowships from the Council of Scientific & Industrial Research (CSIR), Govt. of India.

# Additional information

## Funding

| Funder | Grant reference number | Author |
|---|---|---|
| Science and Engineering Research Board | CRG/2018/001218 | Girish S Ratnaparkhi |
| Pratiksha Trust | EMSTAR/2023/SL03 | Girish S Ratnaparkhi |
| Scheme for Promotion of Academic and Research Collaboration | SPARC-1587 | Girish Deshpande Girish S Ratnaparkhi |

The funders had no role in study design, data collection and interpretation, or the decision to submit the work for publication.

## Author contributions

Subhradip Das, Sushmitha Hegde, Data curation, Formal analysis, Validation, Investigation, Methodology, Writing - original draft, Writing - review and editing; Neel Wagh, Data curation, Formal analysis, Investigation; Jyothish Sudhakaran, Adheena Elsa Roy, Investigation, Methodology; Girish Deshpande, Conceptualization, Formal analysis, Supervision, Funding acquisition, Visualization, Methodology, Writing - original draft, Writing - review and editing; Girish S Ratnaparkhi, Conceptualization, Resources, Formal analysis, Supervision, Funding acquisition, Visualization, Methodology, Writing - original draft, Project administration, Writing - review and editing

## Author ORCIDs

Subhradip Das ⓘ https://orcid.org/0009-0002-4553-966X
Sushmitha Hegde ⓘ https://orcid.org/0000-0003-0777-962X
Girish S Ratnaparkhi ⓘ https://orcid.org/0000-0001-7615-3140

Reviewer #1 (Public review): https://doi.org/10.7554/eLife.98584.3.sa1
Reviewer #2 (Public review): https://doi.org/10.7554/eLife.98584.3.sa2
Reviewer #3 (Public review): https://doi.org/10.7554/eLife.98584.3.sa3
Author response https://doi.org/10.7554/eLife.98584.3.sa4

# Additional files

## Supplementary files

• MDAR checklist

## Data availability

All data generated or analysed during this study are included (1) in the manuscript and supplements, including source data for western blots and (2) additional raw data for images, mass spectrometry, and DNA sequencing files are uploaded to Dryad.

The following dataset was generated:

| Author(s) | Year | Dataset title | Dataset URL | Database and Identifier |
|---|---|---|---|---|
| Das S, Hegde S, Wagh N, Sudhakaran J, Roy AE, Deshpande G, Ratnaparkhi GS | 2024 | Caspar specifies primordial germ cell count and identity in *Drosophila melanogaster* (Supplementary Data) | https://doi.org/10.5061/dryad.zs7h44jkf | Dryad Digital Repository, 10.5061/dryad.zs7h44jkf |

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
