## [Editor Report · eLife Assessment]

This study investigates the role of Caspar (Casp), an orthologue of human Fas-associated factor-1, in regulating the number of primordial germ cells that form during *Drosophila* embryogenesis. The findings are **important** in that they reveal an additional pathway that contributes to germ cell specification and maintenance. The evidence supporting the conclusions is **solid**, as the authors identify Casp and its binding partner Transitional endoplasmic reticulum 94 (TER94) as factors that influence germ cell numbers.

---

## [Referee Report · Reviewer #1 (Public review)]

Summary:

The authors were seeking to define the roles of the *Drosophila* caspar gene in embryonic development and primordial germ cell (PGC) formation. They demonstrate that PGC number, and the distribution of the germ cell determinant Oskar, change as a result of changes in caspar expression; reduction of caspar reduces PGC number and the domain of Oskar protein expression, while overexpression of caspar does the reverse. They also observe defects in syncytial nuclear divisions in embryos produced from caspar mutant mothers. Previous work from the same group demonstrated that Caspar protein interacts with two partners, TER94 and Vap33. In this paper, they show that maternal knockdown of TER94 results in embryonic lethality and some overlap of phenotypes with reduction of caspar, supporting the idea may work together in their developmental roles. The authors propose models for how Caspar might carry out its developmental functions. The most specific of these is that Caspar and its partners might regulate oskar mRNA stability by recruiting ubiquitin to the translational regulator Smaug.

Strengths:

The work identifies a new factor that is involved in PGC specification and points toward an additional pathway that may be involved in establishing and maintaining an appropriate distribution of Oskar at the posterior pole of the embryo. It also ties together earlier observations about the presence of TER94 in the pole plasm that have not heretofore been linked to a function.

Weaknesses:

(1) A PiggyBac insertion allele casp[c04227] is used throughout the paper and referred to as a loss-of-function allele (casp[lof]). While the authors avoid the terms 'null' or 'amorph' and on one occasion refer to the allele as a 'strong hypomorph', nevertheless terming it a 'loss-of-function' allele is misleading. This is because the phenotype of the allele when homozygous is different from the phenotype produced when heterozygous over a deficiency.

(2) The peptide counts in the mass spectrometry experiment aimed at finding protein partners for Casp are extremely low, except for Casp itself and TER94. Peptide counts of 1-2 seem to me to be of questionable significance.

(3) The pole bud phenotypes from TER94 knockdown and casp mutant shown in Fig 5 appear to be quite different. These differences are unexplained and seem inconsistent with the model proposed that the two proteins work in a common pathway. Whole embryos should also be shown, as the TER94 KD phenotype could result from a more general dysmorphism.

(4) Fig 6 is not quantitative, lacking even a second control staining to check for intensity variation artifacts. Therefore it shows that the distribution of Oskar protein changes in the various genotypes, but not convincingly that the level of Oskar changes as the paper claims.

(5) The error bars are huge in the graphs in Fig 7H, I, and J, and in fact these changes are not statistically significant. Therefore the conclusion that 'Reduction in Casp activity specifically affects Smaug degradation during the MZT' is not supported by the data in this figure.

---

## [Referee Report · Reviewer #2 (Public review)]

Summary:

This study investigated the role of the Caspar (Casp) gene, a *Drosophila* homolog of human Fas-associated factor-1. It revealed that maternal loss of Casp led to centrosomal and cytoskeletal abnormalities during nuclear cycles in Drosophila early embryogenesis, resulting in defective gastrulation. Moreover, Casp regulates PGC numbers, likely by regulating the levels of Smaug and then Oskar. They demonstrate that Casp protein levels are linearly correlated to the PGC number. The partner protein TER94, an ER protein, shows similar but slightly distinct phenotypes. Based on the deletion mutant analysis, TER94 seems functionally relevant for the observed Casp phenotype. Additionally, it is likely involved in regulating protein degradation during PGC specification.

Strengths:

This paper uncovers a new function of the Casper (Casp) gene, previously known for its role in immune response regulation and NF-kB signaling inhibition. This new function includes nuclear division and PGC formation in early fly embryos. The findings provide crucial insights into how this pathway contributes to the proper establishment of both somatic cells and the germline, particularly in the context of early embryogenesis. This research is therefore of significant interest to cell and developmental biologists.

Future Research:

While this study has made significant strides in understanding the role of the Casp gene in early embryogenesis, the functional relationships among molecules shown here (Casp, TER94, Osk) and other genes previously known to regulate these processes remain unclear. This underscores the need for future studies to delve deeper into these relationships and their implications.

---

## [Referee Report · Reviewer #3 (Public review)]

Summary:

Das et al. discovered a maternal role for Caspar (Casp), the *Drosophila* orthologue of human Fas-associated factor-1 (FAF1), in embryonic development and germ cell formation. They find that Casp interacts with Transitional endoplasmic reticulum 94 (TER94). Loss of Casp or TER94 leads to partial embryonic lethality, correlated with aberrant centrosome behavior and cytoskeletal abnormalities. This suggests that Casp, along with TER94, promotes embryonic development through a still unidentified mechanism. They also find that Casp regulates germ cell number by controlling a key determinant of germ cell formation, Oskar, through its negative regulator, Smaug.

Strengths:

Overall, the experiments are well-conducted, and the conclusions of this paper are mostly well-supported by data.

Weaknesses:

Some additional controls could be included, and the language could be clarified for accuracy.

---

## [Author Response]

The following is the authors’ response to the original reviews.

This study investigates the role of Caspar (Casp), an orthologue of human Fas- associated factor-1, in regulating the number of primordial germ cells that form during *Drosophila* embryogenesis. The findings are important in that they reveal an additional pathway involved in germ cell specification and maintenance. The evidence supporting the conclusions is solid, as the authors identify Casp and its binding partner Transitional endoplasmic reticulum 94 (TER94) as factors that influence germ cell numbers. Minor changes to the title, text, and experimental design are recommended.

We thank the Editors and Reviewers for their overall positive and thoughtful feedback. Based on these comments, we have revised our manuscript. The changes in the manuscript have been highlighted in ‘blue’ font for easy visualisation.

**Reviewer #1 (Public Review):**
Summary:The authors were seeking to define the roles of the *Drosophila* caspar gene in embryonic development and primordial germ cell (PGC) formation. They demonstrate that PGC number, and the distribution of the germ cell determinant Oskar, change as a result of changes in caspar expression; reduction of caspar reduces PGC number and the domain of Oskar protein expression, while overexpression of caspar does the reverse. They also observe defects in syncytial nuclear divisions in embryos produced from caspar mutant mothers. Previous work from the same group demonstrated that Caspar protein interacts with two partners, TER94 and Vap33. In this paper, they show that maternal knockdown of TER94 results in embryonic lethality and some overlap of phenotypes with reduction of caspar, supporting the idea may work together in their developmental roles. The authors propose models for how Caspar might carry out its developmental functions. The most specific of these is that Caspar and its partners might regulate oskar mRNA stability by recruiting ubiquitin to the translational regulator Smaug.Strengths:The work identifies a new factor that is involved in PGC specification and points toward an additional pathway that may be involved in establishing and maintaining an appropriate distribution of Oskar at the posterior pole of the embryo. It also ties together earlier observations about the presence of TER94 in the pole plasm that have not heretofore been linked to a function.Weaknesses:(1) A PiggyBac insertion allele casp[c04227] is used throughout the paper and referred to as a loss-of-function allele (casp[lof]). However, this allele does not appear to act strictly as a loss-of-function. Figure 1E shows that some residual Casp protein is present in early embryos produced by casp[lof]/Df females, and this protein is presumably functional as the PiggyBac insertion does not affect the coding region. Also, Figures 1B and 1C show that the phenotypes of casp[lof] homozygotes and casp[lof]/Df are not the same; surprisingly, the homozygous phenotypes are more severe. These observations are unexplained and inconsistent with the insertion being simply a loss-of-function allele. Might there be a second-site mutation in casp[c04227]?

The term loss-of-function (lof) is used rather than null or amorph. *casplof* is a strong hypomorph, with residual (and functional) protein estimated in the range 5-10% when compared to the wild type. The *caspc04227* was procured from BDSC, and based on the decrease in lethality of the *casplof/casp(Df)* compared to *casplof*, we assume that second site hits in the *casplof* line are the reason for the enhanced lethality. For this very reason, we have used *casplof/ casp(Df)* for all subsequent experiments. We also conducted rescue experiments wherever possible to confirm the specificity with *caspWT* and various deletion variants of *casp*.

(2) TER94 knockdown phenotypes have been previously published (Zhang et al 2018 PMID 30012668), and their effects on embryonic viability and syncytial mitotic divisions were described there. This paper is inappropriately not cited, and the data in Figure 4 should be presented in the context of what has been published before.

We apologize for the oversight. Indeed, Zhang *et al.* (2018) highlighted TER94 as one of the loci uncovered in their screen and some of the relevant phenotypes are described there. We have referred to their findings at the appropriate junctures as suggested (pg 11, pg13, pg 15).

(3) The peptide counts in the mass spectrometry experiment aimed at finding protein partners for Casp are extremely low, except for Casp itself and TER94. Peptide counts of 1-2 seem to me to be of questionable significance.

Peptide counts are indeed low, but the fact that they are enriched at all, in comparison to controls, considering that we are using whole embryo lysates rather than isolated PGC lysates, suggests interaction with Casp could be biologically/ functionally meaningful. The data is restricted to the supplementary material and is not analyzed in isolation; we have combined data from multiple mass spectrometry experiments by other researchers to link Casp to pole plasm components.

(4) The pole bud phenotypes from TER94 knockdown and casp mutant shown in Fig 5 appear to be quite different. These differences are unexplained and seem inconsistent with the model proposed that the two proteins work in a common pathway. Whole embryos should also be shown, as the TER94 KD phenotype could result from a more general dysmorphism.

We agree that TER94 KD is a stronger phenotype, with TER94 having essential cell division and patterning roles. In fact, the *TER94* RNAi embryos, unlike *casplof*, stall in terms of their developmental program before Stage 4. This has been noted in the earlier study (Zhang et al., 2018). As a result, we focused on pole bud stage embryos that were rare - but present in the collections. We report that PGC from very early *TER94 RNAi* embryos have fewer pole buds.

The rationale behind the presumption that these two proteins may work in a common pathway is clear-cut. We have validated the physical interaction using protein lysates from two developmental time points. Satisfyingly, an affinity purification using antibodies against TER94 or Casp invariably enriches the other protein as the primary interacting partner. Our model integrates data from mammalian and fly systems to support the idea that there must be an overlap between TER94/Casp function, with these two proteins working together to engineer the degradation of ubiquitinated Smaug. Future experiments are necessary to confirm and extend this claim.

(5) Figure 6 is not quantitative, lacking even a second control staining to check for intensity variation artifacts. Therefore, it shows that the distribution of Oskar protein changes in the various genotypes, but not convincingly that the level of Oskar changes as the paper claims.

We appreciate that *oskar* RNA localization is also somewhat altered due to change in *casp* levels. We have acknowledged the variability in the various phenotypes, and as such, it is unsurprising that it has also reflected in the Oskar levels. However, it is evident that a statistically significant number of mutant embryos show a decrease in Oskar levels.

(6) The error bars are huge in the graphs in Figure 7H, I, and J, leading me to question whether these changes are statistically significant. Calculations of statistical significance are missing from these graphs and need to be added.

The data in the Western blots represents the whole embryo, as the lysates used are from embryos 0-1, 1-2, 2-3 hrs. We have averaged and plotted data from 5 Western blots. The changes are not statistically significant. Even without the statistical significance, the data for Fig. 7I led us to examine Smaug in the pole cells, rather than in the whole embryo. The pole cell data (Fig8-D3) is striking and led to the conclusion – that Smaug protein perdures in the pole cells during the stages of syncytial/cellular blastoderm.

(7) There are many instances of fuzzy and confusing language when describing casp phenotypes. For example, on lines 211-212, it is stated that 'casp[lof] adults are only partially homozygous viable as ~70% embryos laid by the homozygous mutant females failed to hatch into larvae'. Isn't this more accurately described as 'casp[c04227] is a maternal-effect lethal allele with incomplete penetrance'? Another example is on line 1165, what exactly is a 'semi-vital function'?

We thank the reviewer for reading the manuscript in detail. We have tried to pay attention to reduce the ambiguity and fixed the text accordingly (pg 7, line 214; pg 33, line 1169, word semi-vital is deleted).

**Reviewer #2 (Public Review):**
Summary:This study investigated the role of the Caspar (Casp) gene, a *Drosophila* homolog of human Fas-associated factor-1. It revealed that maternal loss of Casp led to centrosomal and cytoskeletal abnormalities during nuclear cycles in Drosophila early embryogenesis, resulting in defective gastrulation. Moreover, Casp regulates PGC numbers, likely by regulating the levels of Smaug and then Oskar. They demonstrate that Casp protein levels are linearly correlated to the PGC number. The partner protein TER94, an ER protein, shows similar but slightly distinct phenotypes. Based on the deletion mutant analysis, TER94 seems functionally relevant for the observed Casp phenotype. Additionally, it is likely involved in regulating protein degradation during PGC specification.Strengths:The paper reveals an unexpected function of the maternally produced Casp gene, previously implicated in immune response regulation and NF-kB signaling inhibition, in nuclear division and PGC formation in early fly embryos. Experiments are properly conducted and strongly support the conclusion. The rescue experiment using deletion mutant form is particularly informative as it suggests the requirement of each domain function.Weaknesses:Functional relationships among molecules shown here (and other genes known to regulate these processes) are still unclear.

We completely agree with this assessment. In our view this is an interesting albeit initial report. We also appreciate that understanding the mechanistic underpinnings of these results will be critical. We have ensured that our present claims are backed up by data, however, are fully sensitive to the fact that newer observations will refine or even alter these claims. We are continuing to work on the problem and will hopefully make further inroads in mechanism in the coming years.

**Reviewer #3 (Public Review):**
Summary:Das et al. discovered a maternal role for Caspar (Casp), the *Drosophila* orthologue of human Fas-associated factor-1 (FAF1), in embryonic development and germ cell formation. They find that Casp interacts with Transitional endoplasmic reticulum 94 (TER94). Loss of Casp or TER94 leads to partial embryonic lethality, correlated with aberrant centrosome behavior and cytoskeletal abnormalities. This suggests that Casp, along with TER94, promotes embryonic development through a still unidentified mechanism. They also find that Casp regulates germ cell number by controlling a key determinant of germ cell formation, Oskar, through its negative regulator, Smaug.Strengths:Overall, the experiments are well-conducted, and the conclusions of this paper are mostly well-supported by data.Weaknesses:Some additional controls could be included, and the language could be clarified for accuracy.
**Reviewer #1 (Recommendations For The Authors):**
(1) The paper is inconsistent in using standard *Drosophila* nomenclature. Often the name of the mammalian counterpart is used instead. This needs to be cleaned up as it is very confusing to the reader.

The names of the mammalian counterpart are explicitly used, when we intended, to underscore the parallels between mammalian vs *Drosophila* function, specifically in the context of the major players in this study, TER94 vs VCP; Caspar vs FAF1. Since we do not have direct biochemical data indicating that TER94/Casp degrades Smaug, we use published mammalian literature to draw parallels. At no point have we swapped terminology casually.

(2) The Discussion is far too long and in my view extends too far beyond the experimental data in the paper. As a start for editing, its first two paragraphs (lines 1138-1164) include mostly general statements and could be greatly reduced or eliminated.

Our aim was to emphasize the repurposing of factors between early development and later/adult stages for different functional contexts. Our laboratory (Ratnaparkhi) works on Casp in terms of its roles in NF-kappa B signalling. We serendipitously stumbled on the embryonic lethality while characterizing the *casplof* allele, which, later, led us to examine the function of Casp during embryonic germ cell development.

(3) The Introduction is weak in its description of the developmental function of Toll and Dorsal. This could be summarized in a sentence or two.

As suggested, a few sentences that highlight the developmental function of Toll/Dorsal signalling have been added to the text (pg 3, line 90-92).

(4) Even if correctly cited, it is not appropriate to simply reproduce an image from a public database, as was done in Figure S1C. This should be removed.

Figure S1C has been deleted.

(5) The Materials and Methods section should be moved to after the Discussion so it does not interrupt the flow of the Results.

The Section has been moved as suggested.

**Reviewer #2 (Recommendations For The Authors):**
For general readers, more detailed information about the PGC specification will be helpful in the Introduction or Results section.

PGC specification is introduced in the text as the story transits from global embryonic effects of *casp* knockdown to specific effects on PGCs. A few additional sentences have been added to bolster the text (pg 11, first paragraph).

The Methods section talks about live imaging, but I could not find the experiments in the figures. Are the data available for asynchronous nuclear divisions in the live imaging?

The live imaging relates to DIC movies that are part of Suppl. Fig 2A. The movies are embedded in an MS PowerPoint slide, which has been uploaded as a PowerPoint (and not a PDF).

To ensure that the mutant changes the Osk translation rate, showing the Osk RNA level may be helpful.

*oskar* RNA localization is quite distinct as compared to Oskar and Vasa protein. It has been shown that *oskar* RNA is localized to the founder granules and is, in fact, excluded from the germ granules that contain Vasa, Oskar and *nos* RNA etc. Gavis lab recently reported (Eichler *et al.*, 2020) that ectopic localization of *osk* RNA in the germ granules is toxic to pole cells. Thus, it will be of interest to analyze whether and how *oskar* RNA is localized in *casp* embryos.

More discussion about the difference between Casp and ter94 phenotypes and potential reasons would be informative.

TER94 appears to be an essential maternal gene. Hypomorphic knockdown of TER94 using RNAi is sufficient to induce early embryonic lethality. In fact, Zhang *et. al.*, 2018 et al., using stronger/earlier maternal drivers highlighted the lethality and somatic cell division defects caused due to the severe loss of TER94. The UBX domain is present in multiple proteins, in addition to Casp. TER94 possible plays a vital role in protein degradation of critical cell cycle proteins, such as cyclins that need to be degraded for efficient genomic duplications in the 10’ nuclear division cycles that predominate the first few hours of embryogenesis.

N = 3 (Fig1 legend) and N = 15 (Fig2). What are those numbers?

N = 3 indicated the number of repeats of the western blot. This reference has been deleted. N = 15, represents the number of embryos imaged for data in panels G and H.

**Reviewer #3 (Recommendations For The Authors):**
Major Suggestion:(1) Oskar (Osk) mRNA Localization: Does Osk mRNA localization change upon overexpression or LOF of Casp? Since TER94 has been implicated in Osk mRNA localization (Ruden et al., 2000), this would be a good control to include.

As mentioned earlier, in the response to editors, data presented in our manuscript indicates that Caspar is unique in its ability to regulate both Oskar levels and centrosome dynamics. As the reviewer pointed out, we are in the process of analysing the possible localization defects in *oskar* mRNA in the embryos. Since the preliminary data are promising, we are pursuing this carefully to better understand the involvement of Caspar. We are focusing on the ability of Caspar to regulate early nuclear divisions prior to pole cell formation. It is possible that in *casp* mutant embryos the nuclei/centrosomes that enter the pole plasm are already defective and thus can influence release of the pole plasm components. This needs to be examined carefully, and we are conducting these experiments.

(2) Western Blot for Osk Protein: It would also be beneficial to perform a western blot for Osk protein to demonstrate that it is indeed increased upon Casp overexpression.

This is a good suggestion. However, Oskar antibodies are not readily available, and we have a very limited supply which have been used for embryo staining experiments. We considered these more useful as in addition to the absolute levels, staining experiment can reveal localization pattern. It was thus possible to correlate Oskar function with the pole cell counts in respective genetic backgrounds.

(3) Title Clarification: The title states, "Caspar determines primordial germ cell identity in *Drosophila melanogaster*." The current experiments do not show that Casp determines germ cell identity. It would be more accurate to conclude that Casp regulates germ cell numbers.

Please refer to the introductory paragraphs where we explain our views in this regard. We have modified our title to “Caspar specifies primordial germ cell count and identity in *Drosophila melanogaster*."

Minor Suggestions:(1) Line 69: Delete the use of "recent" for papers published in 2001 and 2007. These papers are around 20 years old.

The word has been deleted.

(2) Paragraph from Line 110: Consider splitting this paragraph into two for better readability and clarity.

Paragraph has been split into two; this has improved readability.

(3) Line 266: Check and correct the formatting issues in this line.

Edited, based on suggestion. A line break was added after the title.

(4) Line 328: Adding references to earlier studies here will be useful for providing context and supporting information.

References that introduce Centrosomes and their roles as organizing centres have been added in line 336.

(5) Line 564: It is best to avoid using the word "master." Please consider using other terms such as "key" or "principal."

Edited, based on suggestion.

(6) Citations: The authors should also cite Cinalli et al., 2013 for the Gcl reference to ensure comprehensive citation of relevant literature.

Thank you for the suggestion. The reference has been added on pages 16 and 29.

(7) Overall Length: The paper is quite long. If it can be shortened, it will be easier to read. Consider condensing sections where possible without losing essential information.

The paper is indeed longer than average, but the choice of *eLife* as the home for this study was, in part, determined by the platform's flexibility regarding length/ word count. It seemed worthwhile to elaborate the text in places to accentuate the novelty of the findings.

These additions and adjustments would help to further substantiate the claims and improve the clarity of the paper.

We hope that the claims made in our manuscript are substantiated by the data that are presented. Wherever possible, we have tried to modify the text suitably to improve clarity.